# Encoding Human Behavior in Information Design through Deep Learning

**Guanghui Yu**
Washington University in St. Louis
guanghuiyu@wustl.edu

**Wei Tang**
Columbia University
wt2359@columbia.edu

**Saumik Narayanan**
Washington University in St. Louis
saumik@wustl.edu

**Chien-Ju Ho**
Washington University in St. Louis
chienju.ho@wustl.edu

## Abstract

We initiate the study of *behavioral information design* through deep learning. In information design, a *sender* aims to persuade a *receiver* to take certain actions by strategically revealing information. We address scenarios in which the receiver might exhibit different behavior patterns other than the standard Bayesian rational assumption. We propose HAIDNet, a neural-network-based optimization framework for information design that can adapt to multiple representations of human behavior. Through extensive simulation, we show that HAIDNet can not only recover information policies that are near-optimal compared with known analytical solutions, but also can extend to designing information policies for settings that are computationally challenging (e.g., when there are multiple receivers) or for settings where there are no known solutions in general (e.g., when the receiver behavior does not follow the Bayesian rational assumption). We also conduct real-world human-subject experiments and demonstrate that our framework can capture human behavior from data and lead to more effective information policy for real-world human receivers.

## 1 Introduction

The problem of information design, where a player with an information advantage (the *sender*) can strategically reveal information to influence another player (the *receiver*) to take certain actions, is ubiquitous in everyday applications. For example, online retailers can highlight a subset of product features to influence buyers to make purchases [32, 46]. Recommendation systems might selectively display other users' ratings to persuade users to follow recommendations [75]. Politicians can influence voters' decisions by designing different policy experiments [1]. There have been various research efforts from economics [51, 62, 29, 30, 44], machine learning and artificial intelligence [79, 12, 2, 33, 6, 25], and general computer science [23, 18] devoted to the study of information design. Among the growing literature on information design, the model of Bayesian persuasion proposed by Kamenica and Gentzkow [39] is one of the most prominent, and has inspired a rich body of studies.

While Bayesian persuasion offers an elegant framework for formulating the information design problem, it has two limitations. First, the receiver is assumed to be Bayesian rational. This means that the receiver can form a posterior in a Bayesian manner and chooses the action that maximizes his expected utility. [1] However, as consistently observed in empirical studies [53, 37], humans often

---

[1] We use she/he to denote the sender/receiver, respectively.

37th Conference on Neural Information Processing Systems (NeurIPS 2023).

deviate from being Bayesian or rational. Directly applying the techniques from the information design literature that assume Bayesian rational receivers could lead to suboptimal outcomes. In this work, we address this limitation by proposing a general optimization framework that can integrate a wide range of human behavior, expressed either as traditional analytical closed-form behavioral models or as data-driven models, and design optimal information policies with respect to the provided human behavior.

Second, despite a decade of effort, characterizing the optimal information policy remains notoriously difficult. Dughmi and Xu [18] have shown that it is #P-hard to compute the optimal expected sender utility, and in multi-receiver settings where each receiver only has binary actions, it is #P-hard to even approximate the optimal sender utility within any constant multiplicative factor [78]. Moreover, most previous works have assumed that the receiver follows the Bayesian rational assumption. When this assumption is relaxed [14, 70], there are generally no known analytical solutions for finding the optimal information policy yet.

In this work, we initiate the study of automated information design that encodes human behavior into the design process. Inspired by the recent effort in utilizing deep learning for auction design [20, 58], we propose HAIDNet, an optimization framework that leverages neural-network architectures for information design. Unlike existing works that assume rational human behavior, our optimization framework can adjust to multiple representations of human behavior patterns, including standard behavioral models represented in analytic forms, and data-driven models trained using machine learning approaches. More specifically, We encode receiver behavior as a function and represent the loss in our optimization framework as a function of the receiver's responses to the disclosed information. This approach enables our optimization framework to accommodate different representations of human behavior and can lead to corresponding optimal information policies. Our contributions can be summarized as follows:

- We initiate the study of automated information design that encodes human behavior in the design process. The proposed end-to-end Human behavior encoded neural network mechanism for Automated Information Design, namely HAIDNet, enables us to optimize the sender's information policy based on a given model of human behavior. We incorporate a neural network architecture for information design problems. Moreover, we extend the literature on deep learning for mechanism design to encode realistic human behavior in the design process.
- We evaluate our approach via extensive simulations. In simpler settings with known analytical solutions, we show that HAIDNet can recover the optimal information policies. We also show that HAIDNet can extend to design information policies for settings that are computationally challenging (e.g., multiple receivers involved), or for settings with no known solutions in general (e.g., when the receiver's behavior does not follow the standard Bayesian rationality assumption).
- Through real-world human-subject experiments, we demonstrate that our framework can adapt to scenarios where we do not have access to human models a priori. We demonstrate that our approach can accurately learn a human descriptor from behavioral data, incorporate it in our optimization framework, and result in more effective information design policy in the real world.

**Related Work.** Our work joins a growing line of research that leverages computational tools for automated mechanism design [11, 65, 10, 66]. More recently, deep neural networks have been utilized for optimizing auction design [20, 27, 31, 13, 42, 61, 58, 45, 12]. Our work differs from this line of work in two ways. First, we extend the approach beyond auction design to address the automated information design problem. Second and more importantly, we have incorporated human behavior in our design, while prior works mostly require standard rationality assumptions.

Our information design formulation builds on the seminal work of Bayesian persuasion [39], which inspired a rich line of research in information design [e.g., see the recent surveys by 38, 4]. In particular, given the practical relevance of information design, there is an increasing number of information design studies in the research community in machine learning and artificial intelligence [79, 12, 2, 33, 25], economics [51, 62, 29, 30, 34], and operations research [44, 76]. Our work differs from most of the existing works in that we integrate human behavior into the design of information policy. There have been a few works addressing non-Bayesian belief updating [14] and non-rational receiver behavior [70, 26] with stylized models in information design. Our work extends previous research by designing an framework that can accommodate both the analytical form and the data-driven form of human behavior. More discussions on related work can be found in Appendix A.

## 2 Preliminary – Bayesian Persuasion Basics

In Bayesian persuasion, there are two players: a sender and a receiver. The sender's goal is to design an information disclosure policy that persuades the receiver to take certain actions maximizing the sender's objective. The state of nature $\theta$ is drawn from a finite set $\Theta \triangleq \{1, \ldots, m\}$ according to a prior distribution $\boldsymbol{\lambda} \triangleq (\lambda(\theta))_{\theta \in \Theta} \in \Delta(\Theta)$. The prior is common knowledge to both the sender and the receiver. The receiver's utility $u^R(a, \theta)$ depends on the receiver action $a \in \mathcal{A}$ from an action set $\mathcal{A}$ and the state $\theta$. The sender's utility $u^S(a, \theta)$ also depends on the receiver's action and the state.

The sender can observe the realized state while the receiver cannot, and the sender can utilize this information advantage to persuade the receiver to take the desired action. In particular, before observing the realized state, the sender can commit to an information policy $\pi$, specifying what signal to present to the receiver conditional on the realized state. More formally, an information policy $\pi$ consists of a signal space $\Sigma$ and a set of conditional probabilities $\{\pi(\cdot|\theta)\}_{\theta \in \Theta}$ where $\pi(\cdot|\theta) = (\pi(\sigma|\theta))_{\sigma \in \Sigma} \in \Delta(\Sigma)$ and $\pi(\sigma|\theta) \in [0, 1]$ denotes the probability to send signal $\sigma \in \Sigma$ given the realized state $\theta$. This information disclosure policy is known to the receiver and specifies how the sender discloses information to the receiver. When a state $\theta \in \Theta$ is realized, the sender sends a signal $\sigma \sim \pi(\cdot|\theta)$ according to the policy.

In Bayesian persuasion, the receiver is assumed to be Bayesian rational in the sense that upon seeing the signal $\sigma$, the receiver forms his posterior belief about the state in a Bayesian manner and takes an action that maximizes his expected utility. Formally, upon seeing the signal realization $\sigma$, the receiver updates his posterior belief over the state of nature, denoted by $\mu(\sigma) \triangleq (\mu(\theta|\sigma))_{\theta \in \Theta} \in \Delta(\Theta)$, by applying Bayes' rule: $\mu(\theta|\sigma) \triangleq \frac{\pi(\sigma|\theta)\lambda(\theta)}{\sum_{\theta' \in \Theta} \pi(\sigma|\theta')\lambda(\theta')}$.

Given the posterior induced from the observed signal $\sigma \in \Sigma$, the receiver takes an action $a^{\mathsf{BR}}(\sigma) \in \mathcal{A}$ that maximizes his expected utility[2], namely, $a^{\mathsf{BR}}(\sigma) \triangleq \arg\max_{a \in \mathcal{A}} \sum_{\theta \in \Theta} \mu(\theta|\sigma) u^R(a, \theta)$. The sender's information design problem is to find the optimal information policy that maximizes her expected payoff induced by the receiver's action, as follows:

$$\max_{\pi} \sum_{\theta \in \Theta} \lambda(\theta) \sum_{\sigma \in \Sigma} \pi(\sigma|\theta) u^S\left(a^{\mathsf{BR}}(\sigma), \theta\right) . \tag{$\mathcal{P}^{\mathsf{BR}}$}$$

In this work, our goal is to design an automated framework to solve the above bi-level optimization problem while encoding realistic human behavior in the design process (i.e., replacing the Bayesian rational human model $a^{\mathsf{BR}}(\sigma)$ with general human behavior).

**Example.** Consider the scenario in which an online retailer (the sender) aims to persuade a buyer (the receiver) to make a purchase. The retailer's products are directly coming from the factory, and the product quality (represented by the binary state $\theta$) is drawn from a prior distribution $\boldsymbol{\lambda}$. The buyer's utility $u^R(a, \theta)$ depends on both his binary purchase decision $a$ and the binary product quality $\theta$, while the retailer's utility $u^S(a, \theta) \equiv u^S(a), \forall \theta$ is state-independent and only depends on the buyer's purchase decision. For example, the goal of the retailer is to persuade the buyer to make a purchase, i.e., $u^S(a) = 1$ for $a = 1$ and $u^S(a) = 0$ for $a = 0$. The buyer only wants to purchase when the product is good, i.e., $u^R(a, \theta) = 1$ if $\theta = a$, and $u^R(a, \theta) = 0$ otherwise.

In order to persuade the buyer to purchase, the retailer can commit to performing (noisy) product inspections $\pi(\sigma|\theta)$ to reveal information about the product quality. For example, the inspection might signal the product quality is satisfactory with $80\%$ chance if the quality of the product is indeed satisfactory (i.e., $\pi(\sigma = 1|\theta = 1) = 0.8$) and signal the product quality is unsatisfactory with $90\%$ chance if the quality is indeed unsatisfactory (i.e., $\pi(\sigma = 0|\theta = 0) = 0.9$). The information design problem for the retailer is to identify an inspection policy that maximizes the probability on selling the product to the buyer.

## 3 HAIDNet: Encoding Human Behavior in Automated Information Design

In this section, we introduce HAIDNet, an optimization framework based on a neural network architecture that can adjust to various forms of human behavior. In the following discussion, we first describe how we modularize human behavior in information design. We then explain the neural

---

[2]BR here stands for Bayesian rational.

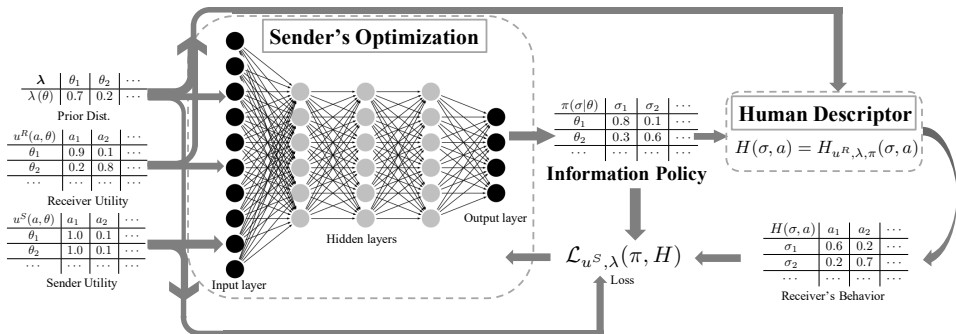

Figure 1: The overall HAIDNet framework. The human descriptor module is given to the optimization module before training. The optimization is performed through back propagation which evaluates the gradient of the loss to update the weights in the neural network structure.

network architecture of our proposed HAIDNet that can adapt to different forms of human behavior. Finally, we outline the procedures for optimizing the information policy in HAIDNet.

## 3.1 Encoding Human Behavior

Bayesian persuasion assumes that the receiver is Bayesian rational. However, in practice, this assumption often does not hold. The receiver may exhibit systematic biases both in belief updating and in decision making. In the following discussion, we formulate the sender's problem on finding the optimal information policy when taking more general human behavior into account.

**Definition 3.1** (Human Behavior Descriptor). For any receiver utility $u^R$, prior $\boldsymbol{\lambda}$, sender information policy $\pi$ (and signal space $\Sigma$), a human behavior descriptor is denoted by $H_{u^R, \boldsymbol{\lambda}, \pi}(\sigma, a)$, representing the probability for a human receiver to choose action $a \in \mathcal{A}$ when seeing a realized signal $\sigma \in \Sigma$.

When the context is clear, we omit the subscripts and write $H_{u^R, \boldsymbol{\lambda}, \pi}(\sigma, a)$ as $H(\sigma, a)$ for notational simplicity. With the above definition, we can rewrite the sender's information design problem as:

$$\max_{\pi} \sum_{\theta \in \Theta} \lambda(\theta) \sum_{\sigma \in \Sigma} \pi(\sigma|\theta) \sum_{a \in \mathcal{A}} H(\sigma, a) u^S(a, \theta) . \qquad (\mathcal{P}^H)$$

Below we give a few examples of human behavior descriptors.

**Bayesian rational (BR).** In standard Bayesian persuasion, the receiver updates his posterior in a Bayesian manner and takes action that maximizes the expected utility. Following the definition in Section 2, the human descriptor can be written as $H(\sigma, a) = \mathbf{1}\{a = a^{\mathsf{BR}}(\sigma)\}$.

**Probability weighting and discrete choice (TH-Model).** We present another human behavior descriptor based on the work by Tang and Ho [70] (denoted as the TH-model in the description of this work). In particular, they combine probability weighting, assuming the receiver's posterior is distorted based on a function $\omega(\cdot) : \Delta(\Theta) \rightarrow \Delta(\Theta)$, and discrete choice model, assuming the receiver's action is stochastic, with a higher probability in taking an action with higher expected utility (based on the distorted posterior belief).

Formally, let $\omega(\theta|\sigma)$ be the receiver's distorted posterior belief after seeing signal $\sigma$ and $\beta_H$ be a parameter in the discrete choice model that tunes how stochastic the receiver's action is (when $\beta_H \rightarrow \infty$, the discrete choice model reduces to standard expected utility maximization), the human behavior descriptor for this model can be written as:

$$H(\sigma, a) = \frac{\exp\left(\beta_H \sum_{\theta \in \Theta} \omega(\theta|\sigma) u^R(a, \theta)\right)}{\sum_{a'} \exp\left(\beta_H \sum_{\theta \in \Theta} \omega(\theta|\sigma) u^R(a', \theta)\right)} . \qquad (1)$$

**Data-driven human behavior descriptor.** Note that in our formulation, we use the function $H(\sigma, a)$ to represent human behavior. Suppose we have access to sufficient human behavioral data, instead of expressing $H(\sigma, a)$ using a closed-form analytical expression, we can train a machine learning model to approximate this function and utilize the learned model as the human behavior descriptor.

## 3.2 HAIDNet Framework and Optimization

We now introduce the framework of HAIDNet and explain how we utilize it to optimize the sender's information policy for a given human descriptor.

**HAIDNet framework.** As presented in Figure 1, HAIDNet consists of two modules: the sender's optimization module and the human descriptor. The sender's optimization module is a neural network responsible for optimizing the sender's optimal information policy. It takes the information design problem instances as input, including the prior distribution $\boldsymbol{\lambda}$ over the states and the payoff functions $u^S, u^R$ for all players. The module outputs an information policy which consists of a set of conditional probabilities $\{\pi(\cdot|\theta)\}_{\theta \in \Theta}$ over the signal space for each state $\theta \in \Theta$.

The human descriptor can either be model-based (e.g., Bayesian rational model or TH model in Equation (1)), or data-driven (e.g., a neural network modeling the receiver's behavior). The human descriptor is treated as a black box from the perspective of the sender's optimization module, and is fixed before HAIDNet begins training. The input of the descriptor consists of the receiver utility $u^R$, the prior distribution $\boldsymbol{\lambda}$, and the information policy $\pi$ (i.e. the output of the sender's optimization module), while the output is the receiver's response strategy $H(\sigma, a) = H_{u^R, \boldsymbol{\lambda}, \pi}(\sigma, a)$.

**Optimization procedure.** For the sender's optimization, we follow the recent line of research on using deep learning for auction design [20]: we randomly draw problem instances from a pre-specified distribution and perform stochastic gradient descent to minimize the loss function in the training process. The loss function is defined to be the negative of the sender's expected utility, since the goal of the sender is to find the optimal information policy that maximizes her expected utility.

$$\mathcal{L}_{u^S, \lambda}(\pi, H) = -\sum_{\theta \in \Theta} \lambda(\theta) \sum_{\sigma \in \Sigma} \pi(\sigma|\theta) \sum_{a \in \mathcal{A}} H(\sigma, a) u^S(a, \theta) \ . \tag{2}$$

Our work differs from previous works in that we incorporate the human behavior descriptor in the definition of the loss function. The requirement is that the human descriptor $H(\sigma, a)$ needs to be differentiable. This requirement is naturally satisfied in many cases, e.g., when the human descriptor follows the model defined in (1) or is a neural-network-based model, the gradient always exists. However, in the Bayesian rational model, since the receiver chooses the action that maximizes his expected utility, this *argmax* operation makes the human model not differentiable. To overcome this issue, we approximate the Bayesian rational model using softmax instead of argmax with a sufficiently large softmax scale parameter $\beta$. [3] More concretely, let $u(a)$ be the expected utility for action $a$. The softmax operator approximates the receiver's behavior by using $\exp(\beta u(a)) / \sum_{a'} \exp(\beta u(a'))$ to denote the probability of choosing action $a$. As a sanity check, when $\beta \to \infty$, this expression reduces to argmax, choosing the action maximizing the expected utility.

**Optimization implementation.** To optimize HAIDNet, we train a neural network with 3 fully connected layers employing ReLU activation functions and the Adam optimizer. The model is trained on 100 batches of size 1024, for a total of $102,400$ uniformly drawn problem instances (i.e., data points for training). Evaluation of the model is conducted on a test set consisting of 1000 problem instances. The specification of hyperparameters and implementation details are included in the appendix. We have also included the source code in the supplementary materials.

## 4 Experiments

### 4.1 Simulations

We have conducted extensive simulations to evaluate HAIDNet. Our results demonstrate that HAIDNet can find the near-optimal information policy in various settings. Specifically, we show its effectiveness in settings where efficient methods exist to obtain the optimal information policy (Section 4.1.1) and in computationally challenging settings where finding the optimal information policy is difficult (Section 4.1.2). Moreover, even in settings where no known solutions exist in general, HAIDNet can generate information policy with good performance (Section 4.1.3). We have conducted additional simulations, including examining the convergence of the training, investigating

---

[3] The notation $\beta$ here is different from $\beta_H$ used to model human behavior in (1).

the scalability of the approach, accounting for varying number of receivers, comparing with random policy, and examining empirical run-time. Due to space constraints, these additional simulation results are included in Appendix B.1.

### 4.1.1 Settings with efficient solutions

We start our evaluations with a simple setting where there exist efficient solutions to find the optimal policy. In this setting, we leverage the efficient solutions as ground truth to examine whether our approach can also identify the optimal information policy.

In particular, we consider the setting with a single Bayesian rational receiver. In this setting, when there are only two actions available for the receiver and there are only two states, there exists a closed-form characterization of the optimal information policy. When the numbers of actions and states are finite constants, the optimal information policy can still be computed efficiently [39]. Therefore, we can evaluate the performance of our approach by comparing the information policy generated by HAIDNet with the optimal policy.

**Binary actions and binary states.** We first examine the simplest setting with binary actions and binary states (a classical setting in Bayesian persuasion [39]), namely, the action space $\mathcal{A} = \{0, 1\}$ and the state space $\Theta = \{0, 1\}$, and observe whether HAIDNet produces near-optimal information policies. For the sender utility, we adopt a stylized setting where the sender obtains utility $1$ when the receiver takes action $1$ and utility $0$ when the receiver takes action $0$. The receiver aims to take the action that aligns with the true state, i.e., $u^R(0, 1) = u^R(1, 0) = 0$, and we randomly draw each value for $u^R(0, 0)$ and $u^R(1, 1)$ from $[0, 1]$. In plain words, the receiver prefers action $1$ when the state is $1$ and action $0$ when the state is $0$, and the goal of the sender is to persuade the receiver to take action $1$. The prior distribution $\boldsymbol{\lambda}$ is drawn from a Dirichlet distribution. We then simulate data using the setting above and optimize HAIDNet.

We first examine whether the policy generated by HAIDNet matches the known optimal policy. Note that in this simple setting, via revelation principle [39], an information policy can be characterized by two signals, i.e., $\sigma \in \{0, 1\}$, where each signal corresponds to a recommended action. Moreover, in the optimal policy, we have $\pi^*(\sigma = 1|\theta = 1) = 1$, and therefore the optimal policy can be characterized by a single parameter $\pi^*(\sigma = 1|\theta = 0)$. To examine whether HAIDNet generates the same policy as the optimal policy, we compare the value of this parameter on different scenarios.

To showcase our results, we present two settings where we have fixed prior distributions: low prior with $\lambda(\theta = 0) = 0.3$ and medium prior with $\lambda(\theta = 0) = 0.5$. [4] For each prior distribution, we vary the receiver utilities and report the parameter $\pi^*(\sigma = 1|\theta = 0)$ both from the optimal policy and from the output of HAIDNet. As visualized in Figure 2, the policy learned by HAIDNet essentially recovers the optimal information policy in almost all scenarios.

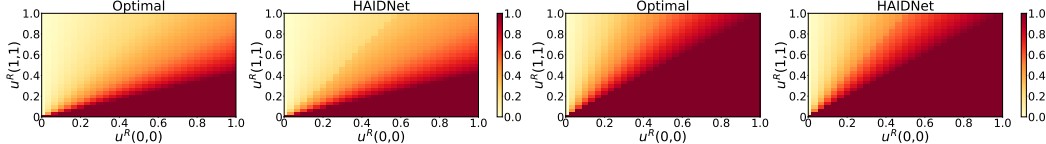

(a) Optimal vs. HAIDNet in the low prior case.  (b) Optimal vs. HAIDNet in the medium prior case.

Figure 2: Comparing the optimal information policy and the policy generated by HAIDNet in the setting with binary actions and binary states.

**Multiple actions and multiple states.** To examine whether our approach scales with the size of the problem instances, we increase the number of states and the number of actions[5]. The performance is measured using the average sender utility. We report both the training performance (e.g., average sender utility for 1,000 instances drawn from instances used for training HAIDNet) and testing performance (e.g., average sender utility for newly drawn 1,000 instances).[6] The results, as shown

---

[4]The results are the same for a wide range of prior distributions.

[5]The results for scaling up both simultaneously are qualitatively the same and are included in the appendix.

[6]We have included additional comparisons to the performance of a simple baseline, random policy, in the appendix. The performance for the random policy is around $0.5$ in all scenarios in this setting.

Table 1: Comparing the average sender utility generated by the optimal policy and the policy from HAIDNet in the setting with a single Bayesian rational receiver.

(a) Increase the number of states $M$ with binary actions. (b) Increase the number of actions $N$ with binary states.

| $M$ | Training | | Testing | | $N$ | Training | | Testing | |
|---|---|---|---|---|---|---|---|---|---|
| | HAIDNet | Optimal | HAIDNet | Optimal | | HAIDNet | Optimal | HAIDNet | Optimal |
| 2 | 0.7409 | 0.7498 | 0.7408 | 0.7451 | 2 | 0.7409 | 0.7498 | 0.7408 | 0.7451 |
| 3 | 0.7737 | 0.7782 | 0.7598 | 0.7669 | 3 | 0.7017 | 0.7214 | 0.7089 | 0.7227 |
| 5 | 0.8171 | 0.8209 | 0.8066 | 0.8225 | 5 | 0.6906 | 0.7113 | 0.6690 | 0.7064 |
| 10 | 0.8495 | 0.8699 | 0.8196 | 0.8686 | 10 | 0.6861 | 0.7084 | 0.6623 | 0.6963 |

Table 2: Comparing the average sender utility generated by the optimal policy and the policy from HAIDNet in the setting with $K$ Bayesian rational receivers.

| $K$ | Training | | Testing | |
|---|---|---|---|---|
| | HAIDNet | Optimal | HAIDNet | Optimal |
| 2 | 0.7887 | 0.7934 | 0.7756 | 0.7873 |
| 3 | 0.7508 | 0.7665 | 0.7379 | 0.7573 |
| 5 | 0.7217 | 0.7458 | 0.7209 | 0.7570 |
| 10 | 0.6971 | 0.7152 | 0.6790 | 0.6966 |
| 15 | 0.6553 | 0.6882 | 0.6621 | 0.6843 |

Table 3: Comparing the average sender utility by the optimal policy and the policy from HAIDNet in the setting with a non-Bayesian-rational receiver parameterized by $\beta_H$.

| $\beta_H$ | Training | | Testing | |
|---|---|---|---|---|
| | HAIDNet | Optimal | HAIDNet | Optimal |
| 1 | 0.5043 | 0.5051 | 0.5041 | 0.5060 |
| 5 | 0.5512 | 0.5557 | 0.5506 | 0.5559 |
| 10 | 0.6045 | 0.6170 | 0.5986 | 0.6168 |
| 50 | 0.7002 | 0.7134 | 0.6800 | 0.7081 |
| 100 | 0.7187 | 0.7291 | 0.6964 | 0.7179 |

in Table 1, demonstrate that our approach works well for large-scale problem instances and also generalizes well to instances not used in training.

### 4.1.2 Settings without efficient solutions

Next, we examine the performance of HAIDNet under the setting where there are no known computationally efficient solutions to characterize the optimal information policy. The goal is to illustrate that HAIDNet performs well even in complicated scenarios and could provide a more efficient approach for settings without analytically tractable solutions.

We consider the setting with multiple receivers and binary actions. The goal is to design a uniform information policy for all receivers (i.e., *public persuasion* [78]). This setting has been shown to be #P-hard to find a policy that approximates the optimal sender utility within any constant multiplicative factor [19]. This means that, unlike the single receiver case, finding the optimal solution for a given problem is practically impossible to solve with a large set of receivers, and we intend HAIDNet to be a new, efficient solver for near-optimal solutions. To examine whether HAIDNet finds the optimal policy, we utilize a brute-force linear-programming approach [19] (the time complexity is exponential in the number of receivers since the number of constraints in the program grows exponentially) to identify the optimal policy when the number of receivers is small. We then compare the information policy generated by HAIDNet and the optimal policy output from the linear programming approach. The receiver utility and prior distributions are generated in the same way as in the single receiver setting. The sender utility is the fraction of receivers choosing action 1, i.e., her utility is given $\frac{|S|}{K}$ if there are $|S|$ receivers choosing action 1 out of a total $K$ receivers.

The simulation results are shown in Table 2. We randomly draw $1,000$ problem instances from the training/testing set and report the average performance of the optimal policy and the HAIDNet policy. As we can see in the results, the performance of the information policy output from HAIDNet is near-optimal. Moreover, HAIDNet provides a much more efficient approach when the number of receivers is large. As a comparison, solving the exact optimal information policy for each problem instance is time-consuming (e.g., it takes more than 23 hours to solve an instance with 18 receivers). On the other hand, HAIDNet only needs to optimize the model once to generate the optimal information policies for all possible problem instances with the same number of receivers (e.g., training HAIDNet with 18 receivers takes slightly more than 1 hour, and generating information policy for a problem instance takes less than 1 second). The empirical run-time comparison is included in the appendix.

### 4.1.3 Settings without known solutions

We now examine the performance of HAIDNet in settings where there are generally no known analytical solutions yet. The goal is to showcase that HAIDNet can be leveraged to address information design problems when we do not have access to solutions.

All our simulations so far have focused on settings which assume that receivers are Bayesian rational. To examine whether HAIDNet works for non-Bayesian-rational receivers, we adopt a relaxation of human behavioral formulation as in Equation (1). While there are no known solutions for identifying the optimal policy in this setting in general, Tang and Ho [70] derived a solution for the simple setting with binary actions and binary states. Therefore, we compare the performance of the optimal policy and the HAIDNet policy in this simple setting under different choices of $\beta_H$ in the human descriptor in Equation (1). Using the same setup as in previous simulations, we report the results in Table 3, showing that HAIDNet works even for a non-Bayesian-rational receiver.

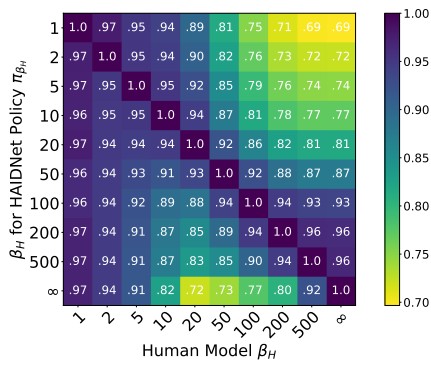

Figure 3: The performance of HAIDNet in settings when the receiver is not Bayesian rational. We train HAIDNet with non-Bayesian-rational receiver model parameterized by $\beta_H$, then evaluate the learned information policy for all receiver models. The performance is normalized so for each human model, the optimal performance is 1.0 among all policies.

Next, we would like to examine how HAIDNet performs in scenarios when there are no known solutions (e.g., in settings with more than binary actions/states). To demonstrate the results, we choose the setting with three states and three actions. The lack of an optimal solution means we cannot evaluate the performance of HAIDNet by comparing its performance with the optimal policy as in the simulations above. Instead, we take a different method and provide evidence to support our approach: We evaluate the set of all learned policies $\pi_{\beta_H}$ against each of the human models $\beta_H$.

For each human model $\beta_H = k$, if $\pi_k$ is the best-performing policy, this indicates that our approach generates a reasonably good information policy. Specifically, for each human model, we compute the performance of each policy available, and we then normalize the set of these performances so that the best-performing performance for each human model has value 1. If our HAIDNet indeed learns a good information policy, we would expect the best performing HAIDNet to be the one trained on the right human descriptor. The results, as shown in Figure 3, demonstrate this behavior and provides evidence that our HAIDNet generates good information policy even when the receiver is not Bayesian rational.

## 4.2 Real-World Human-Subject Experiments

In the simulations, we have assumed access to a closed-form behavior model of the receiver. However, in practice, human behavior is complex and there may not exist a single model that can perfectly represent human behavior. Motivated by this practical concern, we conduct human-subject experiments to examine whether HAIDNet adapts to real-world human behavior. The goal is to examine whether we can utilize data-driven approaches to learn human-behavior descriptors and examine whether HAIDNet performs well when it is paired with data-driven behavior descriptors.

**Task description.** In our human-subject experiments, we present the product purchasing example in Section 2 to human participants. Each human participant is asked to make multiple rounds of purchase decisions. In each round, the participant is presented a product with unknown binary quality (good or bad product). The participant is told that a (noisy) inspection has been performed on the product, and is given the conditional distribution associated with the inspection (i.e., the probability to receive a good/bad signal given the product is good/bad). Finally, the participant is given a realization of the inspection signal and is asked to make a binary decision of purchasing or not. The participant's payment depends on both their purchasing decisions and the true product quality. The task interface is included in Appendix C. The experiment is approved by the IRB in our institution.

Table 4: Test accuracy of different human behavior descriptors in human-subject experiments.

| Model | Bayesian rational | TH-Model | Neural network |
|---|---|---|---|
| Test Accuracy | 0.562 | 0.735 | **0.770** |

**Experiment procedure.** We have recruited 300 workers from Amazon Mechanical Turk. We set the base payment to be $0.50. Workers could earn additional bonuses depending on their performance. The average hourly rate was around $11 USD. The experiment contains two phases as described next.

**Learning human behavior descriptors.** The goal of the first phase is to examine whether we could learn accurate human behavior descriptors from worker's response data. In this phase, we recruited 100 workers, and each worker completed 20 rounds of product purchasing decisions. The parameters of each decision (prior, sender utility, receiver utility, and policy) was drawn uniformly at random. We split the collected data into training/test sets, with 80% of the data for training, and 20% for testing. We trained and examined the performance of three different human behavior descriptors.

- Bayesian rational: This descriptor makes the standard assumption that humans are Bayesian rational. There is no training needed for this descriptor.
- TH-Model: We fit the parameters of the TH model, as described in Section 3.1, from data to minimize the least squares error.
- Neural network: We use a 3 fully connected-layer neural network to fit the data in the training set. We further split the training dataset and use 25% of the data as the validation set to implement early-stopping during training.

We then examine how accurately each descriptor predicts human behavior in the test data. The test accuracy is reported in Table 4. As we can see from the results, the data-driven neural network model leads to the best prediction accuracy, and both TH-Model and the data-driven descriptor significantly outperform the Bayesian rational assumption, reaffirming the need to relax this common assumption.

**Evaluating HAIDNet.** In the second phase, we recruited 200 workers to examine the performance of different information policies. In particular, we examine the following four information policies:

- Random: This information policy is drawn from a Dirichlet distribution.
- BR-policy: The optimal policy when the receiver is a Bayesian rational receiver.
- TH-policy: The optimal policy when the receiver behavior follows the TH-Model, as in Section 3.1.
- HAIDNet: The policy by HAIDNet when we use the neural network learned from the first phase as the human model.

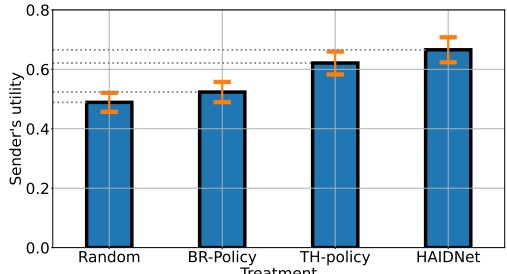

Figure 4: Average sender utility of different policies in human-subject experiments. The differences between BR-policy and TH-policy and between BR-policy and HAIDNet are statistically significant ($p < 0.01$).

When each worker arrives, they are randomly assigned to one of these four policy treatments. They are then presented with 20 rounds of purchase decisions (the parameters of each round are randomly drawn from distributions fixed across all treatments) coupled with the associated information policy in the treatment. We then measure the average sender utility in each treatment. The results, as shown in Figure 4, demonstrate that HAIDNet achieves the best performance. The results showcase the effectiveness of HAIDNet coupled with data-driven human behavior descriptors.

In addition to examining the sender's utility, we also measure the receiver's utility in each treatment (see the full information in Appendix C). We observe that, the policy of HAIDNet leads to an average receiver utility of 0.532, which is the lowest of all four treatments. This creates the concern that when we incorporate the knowledge of receiver behavior to optimize the sender's utility in information design, we are potentially exploiting the knowledge of receiver behavior and hurting the receiver. We offer more discussion on this concern in the discussion section of the paper.

# 5 Conclusion and Discussion

We initiate the study of behavioral information design that encodes human behavior into the design process. We propose HAIDNet, a neural-network-based optimization framework for information design that can adjust to multiple forms of human behavior. Through extensive simulations and human-subject experiments, we demonstrate the effectiveness of HAIDNet in response to different human behavior descriptors. Below we discuss the generalization / limitations and highlight the potential social impacts of this work.

**Generalization and limitations.** While this work has focused on integrating human behavior in automated information design, we believe the methodologies are generalizable to design mechanisms for general human-in-the-loop systems, explicitly encoding realistic human behavior and/or human responses to the system when designing the system. Moreover, our current investigations have adopted the most standard deep learning setup (e.g., full-connected neural networks coupled with stochastic gradient descent). It would be interesting to examine whether the performance could be further improved with carefully crafted network architecture and optimization procedure.

We would like to note the potential limitations of our approach. Our optimization procedure, based on applying stochastic gradient descent on neural networks, does not guarantee to lead to globally optimal solutions in general. Moreover, compared with analytical solutions that are guaranteed to be optimal for all problem instances if the receiver behavior follows the assumption, HAIDNet is a data-driven approach that optimizes the *expected* utility, which requires training data to be representative to ensure generalizability. While our results suggest that HAIDNet recovers the near-optimal policy (e.g., the results in Figure 2), examining the impacts of different training data distributions and whether the results are robust to distributional shifts are potential important future research directions.

Another limitation pertains to the scalability of our proposed approach. While our method exhibits better scalability than exact solvers that utilize linear programming (more detailed discussion is included in Appendix B.1), our current results primarily focus on discrete action/state spaces. As the number of states and actions expands, so does the input size for HAIDNet. It could require much more training iterations to reach convergence. Furthermore, in scenarios with continuous action/state spaces, our approach is not immediately applicable. While discretization might be employed to address the setting with continuous spaces, such an approach requires additional smoothness assumptions to ensure small discretization errors. Overall, understanding and improving the scalability of HAIDNet is an important next step for increasing its practical applicability.

**Potential negative social impacts.** Finally, we highlight the potential negative social implications of the usage of information design frameworks. In information design, the sender often represents the party in power (e.g., the government, social networking platforms), while the receiver is in a less advantageous position (e.g., the general public, users) due to the asymmetry of information access. While it is possible to use information design for social good, guiding the receiver towards actions that are beneficial for himself or the public, the vast majority of information design literature — including our work — focuses on optimizing the sender's utility. When the interests of the sender and receiver are not aligned, optimizing the sender's utility could result in a negative impact on the receivers, who are often the general public. In other words, with an ill-specified objective in information design, the sender could exploit the information advantage and create significant negative social impacts. This concern is further amplified when we obtain more accurate knowledge about the receiver. It is therefore important to consider the impacts and potential regulations on information design.

In light of the concerns raised, to initiate the discussion, we discuss two potential risk mitigation methods. Firstly, on the technical front, we could employ differential privacy techniques [22, 21] to control the amount of private human behavior being incorporated into receiver models. Differential privacy provides a means to balance privacy with utility, typically by introducing controlled noise into the data. This mechanism might be helpful in mitigating the exploitation of marginalized groups, an issue that might be exhibited in our approach. Secondly, from a policy perspective, once we develop a comprehensive understanding of the capabilities of information design with data-driven human models, we, as a society, could and should weigh the utility gains from this method against potential harm. This discussion could then pave the way for the development of regulations and policies for deploying information design. For instance, we might impose constraints ensuring that the deployed information policy does not significantly reduce receiver utility, especially when compared to policies designed assuming standard models such as Bayesian rationality.

## Acknowledgements

We thank the anonymous reviewers for the helpful comments. This work is supported in part by the Office of Naval Research Grant N00014-20-1-2240.

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

# A    Related work

Our work joins a growing number of studies that leverage computational tools for automated mechanism design [11, 65], the problem of utilizing computational approaches or learning-based techniques for finding revenue-maximizing mechanisms in auction settings. One strand of works [10, 66] in this line of research has focused on using learning approaches for mechanism design where only samples of bidder valuations are used to design revenue-maximizing mechanisms. More recently, deep neural networks has been utilized for the automated design of optimal auctions [20], in which the authors propose multiple neural-network architectures for learning approximately optimal auctions. Several works has extended this study in various applications [27, 31, 13, 42, 61, 58, 45, 12]. Our work differs from this line of works in two ways. First, we extend the approach beyond auction design to address the automated information design problem. Second and more importantly, we have incorporated human behavior descriptors in our design, while prior works mostly require standard rationality assumptions.

Our information design formulation builds on top of the seminal work of Bayesian persuasion [39], which initiated a rich theoretical literature on communication games in which a sender can design information to persuade a receiver to take certain actions. Their work has provided theoretical foundations and inspired an active line of research in information design [e.g., see the recent surveys by 38, 4]. Our work builds on top of this line of work through integrating human behavior in the design of information policy, while existing works mostly assume the receiver is Bayesian rational. In particular, our proposed HAIDNet can dynamically adjust to various forms of model-based or data-driven human behavior descriptors. For the model-based receiver behavior, as an example, we have included the probability weighting function [77, 60, 64] for belief updating and the discrete choice model [52, 68, 73] for decision making under uncertainty. Non-Bayesian belief updating in information design also appears in earlier works [14], and the receiver's behavior following the discrete choice model also appears in previous works [70, 26]. Our work generalizes the above in that our framework can adapt to both the above form and the data-driven form of human behavior.

The problem of information design and persuasion has received increasing attention both in research and in practice. For example, researchers have argued that one-quarter of the GDP in the United States is persuasion [51]. Due to its practical relevance, this problem is getting attention more broadly in the general research community, as demonstrated by the recent papers in machine learning and artificial intelligence venues, studying various problem settings such as in security [79], human language interactions [2], data marketplace design [8], algorithmic recourse [33], online recommendation [25], and market competitions [15]. Our work joins this line of study and aims to develop more efficient approaches for information design under more realistic settings of human behavior.

On a conceptual level, our work is related to the growing attention in understanding, modeling, and accounting for human behavior in computational systems, especially in the context of human-robot or human-AI interactions [9, 67, 43, 7, 63, 54, 55, 74]. Moreover, our work joins the recent research theme that incorporates human models in computational and machine learning frameworks [28, 47, 69, 70, 41, 49, 50, 72, 80]. There have been other lines of research that includes humans in the loop of learning frameworks, such as inverse reinforcement learning [56, 24, 67, 36, 81] that infers the reward functions in Markov decision process through (potentially human) demonstrations. Our work differs in that we focused on the information design problem with realistic human receiver models.

Lastly, in this study, we incorporate insights from human behavior into information design. Extensive literature from psychology and behavioral economics has been devoted to deepen our understanding of human behavior. Examples include studies examining deviations from the standard Bayesian assumption in processing information [57, 40, 3] and the rationality assumption in decision-making [37, 52, 68, 73, 35]. While these classical models, often grounded in human data from behavioral experiments [48, 16, 17], offer interpretable behavioral insights, they tend to lack in terms of predictive accuracy. Recently, given the advancements of machine learning techniques and the avaialability of a larger amount of human data, there has been a growing effort to integrate behavioral insights from these classical models with machine learning techniques to enhance predictive accuracy [5, 59]. These models developed in this line of effort are directly applicable in our framework. Moreover, as outlined in Section 5, integrating human behavioral insights into information design can raise concerns about exploiting human irrationality. One potential solution is to incorporate the concept of differential privacy [22, 21, 71]. This would control the amount of personalized information that can be used, preventing undue exploitation.

# B Experimental results and details

In this section, we discuss additional sets of simulation results to highlight the properties and performance of HAIDNet. We also provide details of the optimization process of HAIDNet.

## B.1 Additional experiment results

### B.1.1 Convergence of training

In this set of simulations, we have examined the convergence of training with respect to the number of training iterations and also with respect to the softmax parameter $\beta$ when dealing with Bayesian rational receivers. Overall, HAIDNet converges to finding the optimal policy within reasonable setup.

To illustrate the results, here we present the simplest setting with binary actions and binary states, namely, the action space $\mathcal{A} = \{0, 1\}$ and the state space $\Theta = \{0, 1\}$, and observe whether HAIDNet can produce near-optimal information policies. For the sender utility, we adopt a stylized setting where the sender obtains utility $1$ when the receiver takes action $1$ and utility $0$ when the receiver takes action $0$. We randomly draw each value in the receiver utility $u^R$ from $[0, 1]$. The prior distribution $\boldsymbol{\lambda}$ is drawn from a Dirichlet distribution. We then simulate data using the setting above and optimize HAIDNet.

We compare the performance of the policy learned by HAIDNet with the closed-form optimal policy. Recall that when the receiver is rational (expected utility maximizer), he chooses the action that maximizes his expected utility given his belief about the state. As introduced in Section 3.2, to enable the gradient-based method in optimizing HAIDNet, we replace this *argmax* operation as *softmax* using a softmax scale parameter $\beta$. Therefore, we first examine the impact of this choice of $\beta$ and the amount of training (# iterations in gradient descent) in optimizing the information policy. As shown in Figure 5, when $\beta$ is large enough and when we optimize over a large enough number of data batches, the learned information policy from HAIDNet converges to the information policy that achieves near-optimal performance.

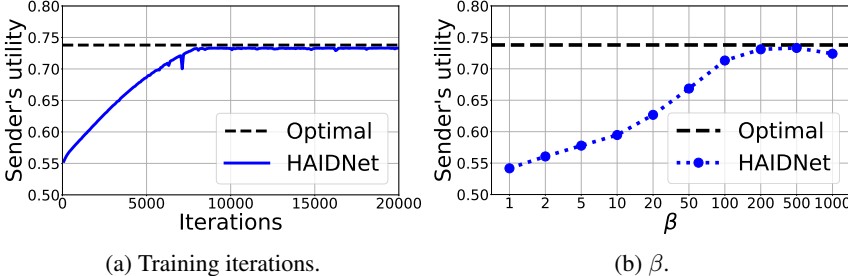

(a) Training iterations.  (b) $\beta$.

Figure 5: The convergence results, with respect to # training iterations and $\beta$, of the sender's utility derived from the information policy generated by HAIDNet.

### B.1.2 Scalability: Empirical run-time comparison

One of the benefits of HAIDNet is to provide efficient solutions for settings when it is computational challenging to derive the optimal policy exactly (e.g., in settings with multiple receivers).

To demonstrate this benefit empirically, we first record the time for computing the exact optimal policy for a problem instance with $K$ receivers via a Linear Programming approach [19]. As we can see from Table 5, the amount of time to compute the optimal information policy grows significantly (the computational complexity grows exponentially as the number of constraints is exponential in the number of receivers in the linear programming approach) as the number of receiver increases. This reaffirms the computational barriers to computing the exact optimal policy. Note that Xu [78] has shown that it is #P-hard to approximate the optimal sender utility within any constant multiplicative factor. So this computational barrier is backed by theoretical analysis.

For HAIDNet, for each class of problems (i.e., a given number of receivers), we only need to train HAIDNet once. For each new problem instance (different priors, sender/receiver utilities, etc), we only need to make a test-time prediction (one pass of forward propagation) to generate an information

policy. Again, in Table 5, we report the time for training HAIDNet and the time for generating the information policy for each problem instance. To provide the number comparisons, when the number of receivers is 18, traditional linear program method of solving the information policy for a problem instance takes more than 23 hours. On the other hand, for HAIDNet, we only need a little more than 1 hour to train HAIDNet for all problem instances with 18 receivers, and it takes less than 1 second to generate the information policy for each receiver. The reported numbers are performed on the machines with Intel(R) Xeon(R) Gold 6148 CPU (2.40GHz) and a Tesla V100-SXM2-32GB GPU.

Table 5: Comparing run-time between HAIDNet and linear programming methods. $K$ is the number of receivers. The reported run-times are in seconds.

| $K$ | Training Time of HAIDNet | Testing time per instance of HAIDNet | Optimal policy per instance via Linear Programming |
|---|---|---|---|
| 2 | 1082 | 0.184 | 0.323 |
| 3 | 1291 | 0.189 | 0.367 |
| 5 | 1571 | 0.221 | 0.371 |
| 10 | 2174 | 0.270 | 4.820 |
| 15 | 3284 | 0.299 | 235.0 |
| 17 | 3713 | 0.333 | 14290 |
| 18 | 4030 | 0.352 | 84280 |

### B.1.3 Additional results for a single Bayesian rational receiver

In Section 4.1.1, we compare the performance between the policy from HAIDNet and the optimal policy in the single Bayesian rational receiver setting with an increasing number of states with binary actions, and an increasing number of actions with binary states. To further complete the results, we have also run simulations when we simultaneously increase the number of actions and the number of states at the same time. To put the performance of HAIDNet into context, we also include the performance of random policy, which provides random signals all the time. This random policy serves as the naive baseline setting. The results are shown in Table 6c. The average sender utility obtained by HAIDNet policy is close to optimal policy in both training and testing problem instances (averaged over 1,000 instances) even in cases with large action and state numbers. We also evaluated the model training error for binary action case and binary state case in Table 6, which shows that HAIDNet works well for large-scale problem instances.

### B.1.4 Additional results for multiple receivers and non-Bayesian-rational receivers

Due to space constraints, we do not include all results in these two settings in the main paper. Here we provide the full results, including the performance of random baseline as well. The results for settings with multiple receivers are included in in Table 7, and the results for settings with a single non-Bayesian-rational receiver are included in Table 8. HAIDNet performs well in multiple receiver settings and non-Bayesian-rational receiver settings.

Recall that for all the results presented in this work, both the training and testing performance are the average performance for $1,000$ data points sampled from the training and testing datasets.

### B.1.5 Varying number of receivers, actions and states

In our main paper, each HAIDNet can accommodate any problem instance (i.e., different specifications of priors, receiver utility, and sender utility) with a fixed number of actions, states, and receivers. It is then natural to wonder whether we can extend the HAIDNet structure so that it can work with varying numbers of receivers, actions, and states. As a proof of concept, in this set of simulations, we attempt to address this question and present an approach that can work with varying numbers of receivers, actions, and states when the numbers are upper bounded.

We first examine the relaxation of a fixed number of receivers. In particular, we can generalize our approach to address varying numbers of receivers when the number of receivers is upper bounded. One straightforward approach is to maintain multiple HAIDNet, one for each fixed number of receivers, for generating the optimal information policy. Another approach is to train a HAIDNet that can generate information policy for the maximum number of receivers. In settings when the

Table 6: Comparing the average sender utility generated by the optimal policy and the policy from HAIDNet in the setting with a single Bayesian rational receiver.

(a) Increase the number of states $M$ with binary actions.

| $M$ | Training | | | Testing | | |
|---|---|---|---|---|---|---|
| | Random | HAIDNet | Optimal | Random | HAIDNet | Optimal |
| 2 | 0.4901 | 0.7409 | 0.7498 | 0.4909 | 0.7408 | 0.7451 |
| 3 | 0.5009 | 0.7737 | 0.7782 | 0.4819 | 0.7598 | 0.7669 |
| 5 | 0.4898 | 0.8171 | 0.8209 | 0.5227 | 0.8066 | 0.8225 |
| 10 | 0.4841 | 0.8495 | 0.8699 | 0.4838 | 0.8196 | 0.8686 |

(b) Increase the number of actions $N$ with binary states.

| $N$ | Training | | | Testing | | |
|---|---|---|---|---|---|---|
| | Random | HAIDNet | Optimal | Random | HAIDNet | Optimal |
| 2 | 0.4901 | 0.7409 | 0.7498 | 0.4909 | 0.7408 | 0.7451 |
| 3 | 0.4911 | 0.7017 | 0.7214 | 0.5064 | 0.7089 | 0.7227 |
| 5 | 0.4919 | 0.6906 | 0.7113 | 0.5119 | 0.6690 | 0.7064 |
| 10 | 0.4907 | 0.6861 | 0.7084 | 0.4861 | 0.6623 | 0.6963 |

(c) Increase both the number of states and actions. $M = N$ represents state number equals action number.

| $M = N$ | Training | | | Testing | | |
|---|---|---|---|---|---|---|
| | Random | HAIDNet | Optimal | Random | HAIDNet | Optimal |
| 2 | 0.4901 | 0.7409 | 0.7498 | 0.4909 | 0.7408 | 0.7451 |
| 3 | 0.4791 | 0.7352 | 0.7679 | 0.4854 | 0.7199 | 0.7587 |
| 5 | 0.5029 | 0.7771 | 0.8113 | 0.5101 | 0.7755 | 0.8121 |
| 10 | 0.4799 | 0.8613 | 0.8971 | 0.4872 | 0.8323 | 0.8994 |
| 50 | 0.4903 | 0.9247 | 0.9550 | 0.7058 | 0.9166 | 0.9545 |

Table 7: Comparing the average sender utility generated by the optimal policy and the policy from HAIDNet in the setting with $K$ Bayesian rational receivers.

| $K$ | Training | | | Testing | | |
|---|---|---|---|---|---|---|
| | Random | HAIDNet | Optimal | Random | HAIDNet | Optimal |
| 2 | 0.5158 | 0.7887 | 0.7934 | 0.5195 | 0.7756 | 0.7873 |
| 3 | 0.5050 | 0.7508 | 0.7665 | 0.4898 | 0.7379 | 0.7573 |
| 5 | 0.4920 | 0.7217 | 0.7458 | 0.4980 | 0.7209 | 0.7570 |
| 10 | 0.5063 | 0.6971 | 0.7152 | 0.5192 | 0.6790 | 0.6966 |
| 15 | 0.5007 | 0.6553 | 0.6882 | 0.4841 | 0.6621 | 0.6843 |
| 17 | 0.5037 | 0.6166 | 0.6503 | 0.5004 | 0.6160 | 0.6497 |

Table 8: Comparing the average sender utility by the optimal policy and the policy from HAIDNet in the setting with a non-Bayesian-rational receiver parameterized by $\beta_H$.

| $\beta_H$ | Training | | | Testing | | |
|---|---|---|---|---|---|---|
| | Random | HAIDNet | Optimal | Random | HAIDNet | Optimal |
| 1 | 0.4986 | 0.5043 | 0.5051 | 0.5006 | 0.5041 | 0.5060 |
| 5 | 0.4929 | 0.5512 | 0.5557 | 0.5036 | 0.5506 | 0.5559 |
| 10 | 0.4904 | 0.6045 | 0.6170 | 0.5064 | 0.5986 | 0.6168 |
| 50 | 0.4919 | 0.7002 | 0.7134 | 0.5093 | 0.6800 | 0.7081 |
| 100 | 0.4931 | 0.7187 | 0.7291 | 0.5097 | 0.6964 | 0.7179 |
| 500 | 0.4946 | 0.7418 | 0.7468 | 0.5096 | 0.7354 | 0.7396 |

number of receivers is less than the maximum number, we can include "null receivers" who always choose action 0 (by setting the receiver utility such that the utility for taking action 0 is always larger than taking other actions in both states). By including this in the training process, we can have a single HAIDNet that can generate policies for a bounded variable number of receivers. As a proof of concept, we have implemented the above approach and trained a HAIDNet that can work with up

to 10 receivers. We then examine its performance when the number of receivers is smaller than 10. As we can see from Table 9, this approach achieves reasonable performance and shows promising results.

Table 9: Comparing the average sender utility by the optimal policy and the policy from HAIDNet in the setting with at most 10 Bayesian rational receivers.

| $K$ | Training | | | Testing | | |
|---|---|---|---|---|---|---|
| | Random | HAIDNet | Optimal | Random | HAIDNet | Optimal |
| 2 | 0.5042 | 0.7830 | 0.8018 | 0.4986 | 0.7538 | 0.7921 |
| 3 | 0.5066 | 0.7337 | 0.7586 | 0.4866 | 0.7139 | 0.7450 |
| 5 | 0.5032 | 0.7245 | 0.7451 | 0.5071 | 0.7121 | 0.7387 |
| 10 | 0.4944 | 0.6911 | 0.7118 | 0.5009 | 0.6650 | 0.6901 |

We now examine whether this approach also works for extending the number of states $M$ and the number of actions $N$. As a proof of concept, we adopt the same approach above and train a HAIDNet for a maximum of 5 actions and 5 states. We then examine the performance of HAIDNet for problem instances with less or equal to 5 actions or states. As shown in Table 10, this approach also works in addressing varying numbers of actions and states.

Table 10: Comparing the average sender utility by the optimal policy and the policy from HAIDNet in the setting with at most 5 states and 5 actions, for single Bayesian rational receivers.

| $(M, N)$ | Training | | | Testing | | |
|---|---|---|---|---|---|---|
| | Random | HAIDNet | Optimal | Random | HAIDNet | Optimal |
| (2, 3) | 0.4994 | 0.6564 | 0.7308 | 0.5276 | 0.6517 | 0.7411 |
| (2, 4) | 0.4852 | 0.6450 | 0.7134 | 0.5236 | 0.6535 | 0.7329 |
| (2, 5) | 0.4898 | 0.6498 | 0.7042 | 0.5111 | 0.6641 | 0.7258 |
| (3, 2) | 0.5094 | 0.6856 | 0.7735 | 0.4731 | 0.6462 | 0.7574 |
| (3, 3) | 0.5128 | 0.7072 | 0.7791 | 0.4832 | 0.6689 | 0.7615 |
| (3, 4) | 0.5343 | 0.7165 | 0.7729 | 0.5322 | 0.6940 | 0.7672 |
| (3, 5) | 0.4798 | 0.6922 | 0.7453 | 0.5308 | 0.6990 | 0.7492 |
| (4, 2) | 0.4898 | 0.6849 | 0.7701 | 0.5216 | 0.6922 | 0.7883 |
| (4, 3) | 0.4721 | 0.7051 | 0.7761 | 0.4796 | 0.6940 | 0.7844 |
| (4, 4) | 0.5032 | 0.7239 | 0.7812 | 0.5143 | 0.7186 | 0.7962 |
| (4, 5) | 0.4700 | 0.7347 | 0.7807 | 0.5144 | 0.7421 | 0.7925 |
| (5, 2) | 0.4883 | 0.7147 | 0.7915 | 0.5186 | 0.7137 | 0.8038 |
| (5, 3) | 0.5394 | 0.7736 | 0.8398 | 0.4928 | 0.7318 | 0.8184 |
| (5, 4) | 0.4998 | 0.7810 | 0.8289 | 0.4951 | 0.7494 | 0.8242 |
| (5, 5) | 0.4819 | 0.7722 | 0.8159 | 0.4863 | 0.7605 | 0.8079 |

## B.2 Data generation

Here we provide the details in generating data instances for training HAIDNet in our settings.

**Single receiver, binary actions and binary states.** In the simplest setting with binary actions and binary states, the action space is $\mathcal{A} = \{0, 1\}$ and the state space is $\Theta = \{0, 1\}$. We adopt a stylized setting for binary actions where the sender obtains utility 1 when the receiver takes action 1 and utility 0 when the receiver takes action 0 [39]. The receiver utility $u^R$ is uniformly drawn from $[0, 1]$ and prior distribution is draw from Dirichlet distribution. We filter out trivial problem instances where the receiver will always choose one action whatever the information policy, e.g., the receiver always chooses action 1 when receiver utility $u^R(1, \theta) > u^R(1, \theta), \forall \theta \in \Theta$. Total 102,400 instances are generated for training, 1,000 for validation and 1,000 for testing.

**Single receiver, multiple actions, and multiple states.** In the setting with $N$ actions and $M$ states, the action space is $\mathcal{A} = \{0, 1, \ldots, N - 1\}$ and the state space is $\Theta = \{0, 1, \ldots, M - 1\}$. The sender utility is set to $u^S(a, \theta) = \frac{a}{N-1}, \forall \theta \in \Theta$ if $N \geq 3$, and the same as above binary actions if $N = 2$. The receiver utility $u^R$ is uniformly drawn from $[0, 1]$ and prior distribution is drawn from Dirichlet distribution. We also filter out trivial cases where the receiver will always choose one action whatever the information policy is.

**Multiple receivers, binary actions, and binary states.** The receiver utility and prior distributions are generated in the same way as in the cases of a single receiver, binary actions, and binary states. The sender utility is the fraction of receivers choosing action 1, i.e., her utility is given $\frac{|S|}{K}$ if there are $|S|$ number of receivers choosing action 1 and $K$ is the total number of receivers. We also filter out trivial cases where the receiver will always choose one action whatever the information policy is.

**Problem instances in human-subject experiments.** In our human-subject experiment, the problem setup is the same as the setting with a single receiver, binary actions, and binary states. To make the setting easier to understand for experiment participants, the receiver utility is drawn from $\{1, 2, 3, 4, 5\}$ when the participant chooses to purchase a good product or chooses to not purchase a bad product, and the participant utility is 0 for other cases. The sender utility is set to 1 when the receiver chooses to buy, and 0 otherwise. The prior distribution is drawn from the Dirichlet distribution, however, we round all probability in the prior distribution and the information policy to the nearest tenth digit, $\{0\%, 10\%, \ldots, 100\%\}$, to make it easier to interpret for human participants.

### B.3  HAIDNet optimization procedures

Here we provide more detailed parameter setups for our simulations and human subject experiments.

**Optimizing HAIDNet in simulations.** We build a 3 fully connected layer neural network with ReLU activation functions as the sender's optimization module in HAIDNet. Network parameters are initialized by Glorot uniform initializer. When optimizing HAIDNet, we use the Adam optimizer and batch gradient descent. Batch size is 1,024, batch number is 100, and maximum training epoch (each epoch contains 100 batches) is 1,000.

The hyperparameters are tuned by using the validation dataset. We then report the performance on the test dataset. The number of nodes for each hidden layers is tuned in the range of $\{64, 128, 256, 512, 1024\}$, and the initial learning rate is tuned in the range of $\{0.001, 0.002, 0.005, 0.01, 0.02, 0.05, 0.1\}$. When the human descriptor is Bayesian rational, we use softmax to smoothen the argmax operator. Empirically we find directly assigning large value to $\beta$ leads to bad performance. So instead, we increase $\beta$ gradually from 10 to 1000 exponentially in first 100 epochs of training, that is $\beta_i = 10^{1+\frac{i}{50}}$ in $i$'th epoch, and maintain $\beta = 1000$ for remaining training process. In the setting with multiple actions and multiple states, softmax approximation of Bayesian rational behavior leads to higher errors. We further adapt the approach of iteratively training networks, reweighting data distributions, and aggregate learned neural networks to reduce the error. Empirically, aggregating three models are enough to reach promising performance.

**Optimizing HAIDNet in human-subject experiments.** After collecting human responses in the first phase of human experiments, we fit TH-Model and train a neural network model for human behavior descriptors. $\beta_H$ in TH-Model is fitted by minimizing a square error between model prediction and human data, and $\beta_H = 20$ fits the best. For the neural network model, we use a 3 fully connected-layer neural network with ReLU activator. We split the data into training/testing sets, with 80% of the data for training, and 20% of the data for testing. We further split the training dataset and use 25% of the training dataset as a validation set to implement early-stopping during training. The neural network for fitting human behavior is trained with batch number 12, batch size 100, and maximum epoch 100. The number of nodes for each hidden layers is tuned in $\{16, 32, 64, 128, 256, 512, 1024\}$ and the initial learning rate is tuned in $\{0.001, 0.002, 0.005, 0.01\}$. We select the hyper-parameter based on average performance of validation sets. Because of 5-fold splitting, we have 5 trained neural networks for human descriptors, and we train HAIDNet corresponding to each of the human model. The learned policy are close in terms of expected utility, in range of $[0.697, 0.726]$, and we select the model of the highest performance in simulation to design the information policy in the second phase experiment.

## C  Details of Human-Subject Experiments

We provide more detailed information about our human-subject experiments here. We compare average sender utility of different policies in human-subject experiment in Figure 4, and we also compute the receiver utility in each treatment, included in Table 11. As we can see from the table, while HAIDNet helps find a policy that leads to the highest sender utility, it comes at the cost of reducing the receiver utility, a demonstration of the ethical concerns as discussed in Section 5.

Table 11: Comparing sender and receiver utility of different policies in human-subject experiments.

| Information Policy | Random | BR-Policy | TH-Policy | HAIDNet |
|---|---|---|---|---|
| Sender Utility | 0.489 | 0.524 | 0.621 | 0.667 |
| Receiver Utility | 0.663 | 0.634 | 0.565 | 0.532 |

In our experiment setup, given the sender's goal is to have the receiver purchase the products regardless of the product quality, when the sender is more successful, it leads to a lower receiver utility in general and implies the potential negative social impacts.

## C.1 Demographic of Workers

We have recruited 300 workers from Amazon Mechanical Turk in total for our experiments. Table 12 contains the demographic information of the 300 workers.

Table 12: Demographic information of the participants in our experiment.

| Group | Category | Number |
|---|---|---|
| Age | 20 to 29 | 88 |
| | 30 to 39 | 111 |
| | 40 to 49 | 65 |
| | 50 to 59 | 25 |
| | 60 or older | 11 |
| Gender | Female | 131 |
| | Male | 168 |
| | Other | 1 |
| Race / Ethnicity | Caucasian | 240 |
| | Black or African-American | 18 |
| | American Indian/Alaskan Native | 5 |
| | Asian or Asian-American | 22 |
| | Spanish/Hispanic | 6 |
| | Other | 9 |
| Education | High school degree | 12 |
| | Some college credit, no degree | 9 |
| | Associate's degree | 24 |
| | Bachelor's degree | 223 |
| | Graduate's degree | 29 |
| | Other | 3 |

## C.2 Task Interface and Description

In our human-subject experiments, we simulate the setting with binary actions and binary states. In particular, we present the product purchasing example as we discussed in Section 2. The task interface about our human-subject experiments is shown in Figure 6.

Each human participant is asked to make multiple rounds of purchase decisions. In each round, the participant is presented a product with unknown binary quality (either good product or bad product). The participant is told that a (noisy) inspection has been performed on the product, and is given the conditional distribution associated with the inspection (i.e., the probability to receive a good/bad signal given the product is good/bad). Finally, the participant is given a realization of the inspection signal and is asked to make a binary decision of purchasing or not. The participant's reward depends on both their purchasing decisions and the true product quality. When collecting human response in the first phase, random policy are presented to all participants. In the second phase, different policies are presented: {Random, BR-policy, TH-policy, HAIDNet policy}. The policies are designed with the assumption that the sender is persuading human receivers to purchase the product, and we calculate the probability of participants choosing to purchase and report it as the sender utility to evaluate performance of different policies.

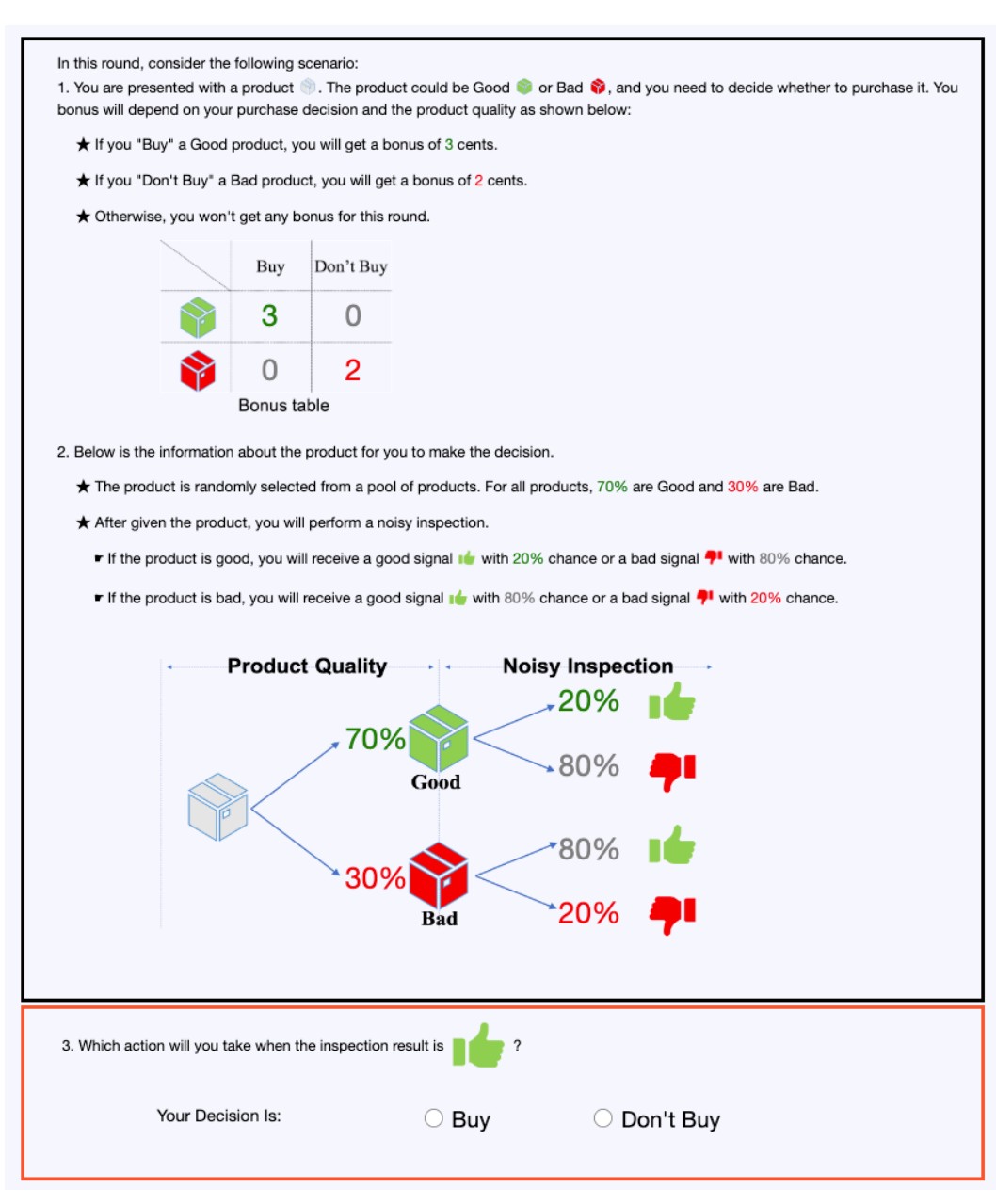

In this round, consider the following scenario:

1. You are presented with a product 📦. The product could be Good 📦 or Bad 📦, and you need to decide whether to purchase it. You bonus will depend on your purchase decision and the product quality as shown below:

★ If you "Buy" a Good product, you will get a bonus of 3 cents.

★ If you "Don't Buy" a Bad product, you will get a bonus of 2 cents.

★ Otherwise, you won't get any bonus for this round.

|  | Buy | Don't Buy |
|---|---|---|
| 📦 | 3 | 0 |
| 📦 | 0 | 2 |

Bonus table

2. Below is the information about the product for you to make the decision.

★ The product is randomly selected from a pool of products. For all products, 70% are Good and 30% are Bad.

★ After given the product, you will perform a noisy inspection.

☞ If the product is good, you will receive a good signal 👍 with 20% chance or a bad signal 👎 with 80% chance.

☞ If the product is bad, you will receive a good signal 👍 with 80% chance or a bad signal 👎 with 20% chance.

**Product Quality**    **Noisy Inspection**

70% Good — 20% 👍 / 80% 👎

30% Bad — 80% 👍 / 20% 👎

3. Which action will you take when the inspection result is 👍 ?

Your Decision Is:    ○ Buy    ○ Don't Buy

Figure 6: Human experiment interface.

