# OpenReview forum: "Encoding Human Behavior in Information Design through Deep Learning"
_NeurIPS.cc/2023/Conference — NeurIPS 2023 poster_

### Official Review · Reviewer_34E5 · 2023-07-07

**Soundness:** 4 excellent
**Presentation:** 3 good
**Contribution:** 2 fair
**Rating:** 4
**Confidence:** 5

**Summary:**

The study demonstrates commendable efforts in employing supervised learning techniques and utilizing Amazon Mechanical Turk to acquire data to develop a human behavior descriptor. The authors further utilize neural networks to optimize the sender's signaling scheme based on the fitted human decision-making model, aiming to maximize their expected returns. The experiments conducted are extensive and encompass a broad range of participants that avoids bias.

**Strengths:**

The work is one of the early works that attempt neural network-based approaches for information design. The work also heavily involves human behavior which is meaningful and potentially useful.

**Weaknesses:**

Major issues:

1. In line 27, the assumption of Bayesian rationality does not appear to be a limitation of previous works. Prior research primarily focused on studying information design problems, such as equilibrium between agents and when senders can benefit from persuasion. The authors' research, however, addresses a different direction, representing a distinct problem. Therefore, the assumption of previous works is not supposed to be claimed as a "limitation".
2. This paper implies that $H$ must have encountered all possible $\pi$, for the two-stage process employed to be reasonable. Otherwise, the sender and receiver should adapt to each other iteratively in the search for equilibrium, rather than being divided into two distinct stages. This is a strong assumption because $\pi$ is continuous. But the paper does not emphasize this assumption adequately. Furthermore, treating continuous variables $\pi$ and priors as features is a simplistic neural approach that may perform worse when the state and action spaces become larger.

Minor comments:

1. This paper explores information design problems in a sequential manner with multiple receivers. From the view of the settings, the receivers are set to be independent and they may have different priors, and the sender will persuade them through a public communication channel. It would be better if the authors emphasize these assumptions. Also, I notice that there are several previous works that study this kind of task, e.g., [Interactive Information Design](https://shs.hal.science/halshs-01791918/file/Publi_2.pdf) and [State-Dependent Central Bank Communication with Heterogeneous Beliefs](https://papers.ssrn.com/sol3/papers.cfm?abstract_id=3923047).
2. In line 54, the author claims to be the first to incorporate neural networks. However, we have come across some arXiv preprints that have already explored this area, such as: [Learning to Persuade] (https://openreview.net/forum?id=0oSM3TC9Z5a).
3. The example in line 110 may require more description. For example, the reward functions of both agents is not formally introduced.
4. In lines 218-223, does the author provide a more detailed description of this example? Why $\pi^*(\sigma=1\mid\theta=1)$? As far as I know, the optimal signaling scheme would be symmetric if the seeds of the experiment codes are random.
5. In line 285, why are there no known solutions for three states and three actions? Shouldn't this be solvable in a similar manner?
6. The human experiments are intriguing, and I am curious about some issues regarding these experiments. For example, during the experiments, did the human participants undergo a learning process? Did the players have any prior experience with trial runs? Were the players allowed to communicate with each other? Also, I noticed that the questionnaires provided in the appendix appear to be quite professional. How did the authors ensure that the participants understood the game?

**Questions:**

N/A

---

> ### Author Rebuttal · Authors · 2023-08-10
>
> We thank the reviewer for the thoughtful review.  We first respond to your major concerns.
>
> **[The assumption of Bayesian rationality as a limitation]** While Bayesian persuasion has provided an elegant framework for studying information design, it has made several assumptions that limit the practical impacts. One of them is the assumption that the receiver is Bayesian rational. Meanwhile, extensive empirical works (e.g., Camerer 1998; Benjamin 2019) have shown that people often systematically depart from Bayes’s rule when confronted with new information. By “limitation” we mean that directly applying the techniques assuming Bayesian rational receivers when the receiver is not would lead to suboptimal outcomes. As also noted in recent work by de Clippel & Xu, JPE’22, new treatments need to be developed to account for receivers who make mistakes in probabilistic inference and decision making. We will make our descriptions more clear to reflect this.
>
> **[Assumptions on human models]** We want to first highlight that our framework is designed to accommodate various representations of human behavior. As shown in our evaluations, it can handle both standard closed-form representations of human models (e.g., Bayesian rationality or TH-model) and data-driven forms.  The reviewer's concern is mostly surrounding the data-driven human models. Indeed, if our human model is entirely trained on historical data, we would require the training data to be representative. Note that even in training data-driven human models, we do not require the data to encounter “all” scenarios, as the goal of ML is to generalize beyond the training data.
>
> We also want to note that this concern can be mitigated by leveraging the recent developments at the intersection of cognitive science and ML that have shown we can utilize established human models (like Bayesian rationality or the TH-model) as priors and harness data to refine these models [Bourgin et al. ICML 2019, Peterson et al. Science 2021]. The main benefit of this approach is that it provides a reasonable model to begin with even when we do not have enough data and could lead to a more accurate model with more data. Our framework can seamlessly integrate with human models developed using such methodologies.
>
> - Bourgin et al. "Cognitive model priors for predicting human decisions." ICML. 2019.
> - Peterson et al. "Using large-scale experiments and machine learning to discover theories of human decision-making." Science. 2021.
>
> We now respond to your minor concerns.
>
> **[Multiple receivers]** There seems to be misunderstandings of our settings. In the multi-receiver setting that we are addressing, our goal is to design a single information policy for multiple (potentially heterogeneous) receivers simultaneously. We do not interact with each receiver in a sequential manner. Our multi-receiver setting is known to be public persuasion [60]. We will provide clarification of our settings and also include discussion to the additional related work as the reviewer suggested.
>
> **[Related work]** Thanks for the pointer! We will include the work in our revisions. We didn’t notice this paper since it doesn't seem to be published yet and doesn’t have an public arXiv version (albeit it has an openreview record). Our work is generally a more comprehensive investigation of automated information design and has addressed general forms of human behavior.
>
> **[Explain the optimal information policy of binary actions/states]** In this example, we adopt the most classical setting with binary states/actions introduced in the original Bayesian persuasion paper [32].  It’s our oversight that we didn’t specify all the details. In particular, the receiver utility $u^R(0,1)=u^R(1,0) = 0$, and we randomly draw values from $[0,1]$ for $u^R(0,0)$ and $u^R(1,1)$. In plain words, in this setting, the receiver prefers action 1 when the state is 1 and action 0 when the state is 0, and the goal of the sender is to persuade the receiver to take action 1.  We will fix the description and provide a full description.
>
> **[No known solutions for non-Bayeisan-rational receivers]** To the best of our knowledge, the studies on deriving optimal information policies when the receiver is not Bayesian rational is limited. The solution for the setting with binary action/states is from Tang and Ho [56]. However, their approach analytically derives a closed-form solution for the optimal information policy for binary actions/states, and their derivation doesn’t seem trivial to extend to multi-state and multi-action settings.
>
> **[Clarifications of human experiments]** The participants cannot discuss with each other, and each individual is allowed to participate in the study only once. During the human experiments, participants were provided with instructions and an illustrative example of the problem instance to help them understand the task setting. We also offered incentives in the form of bonus payments for those who correctly inferred the latent state and made the right decision. However, despite these measures, we cannot guarantee that every participant fully understands the task or puts in a reasonable effort. Nonetheless, this is consistent with many real-world human decision-making scenarios. Furthermore, as mentioned in our response to Reviewer 7rN5, our framework is effective as long as the human behavior used to train the neural network (NN) for the information policy comes from the same distribution as the human behavior observed during deployment. Our experiments aim to meet this condition by recruiting first-time participants from the MTurk platform for both phases.

---

> > ### Comment · Reviewer_34E5 · 2023-08-15
> > **Thank you for the rebuttal**
> >
> > I thank the authors for providing a response. I would focus on the two main issues for the discussion.
> >
> > I believe that the rationality assumption is rather general, and the authors' assumption is instead limited. My argument is as follows: For computational problems, rationality assumption is the most general setting that succinctly describes a fully utilitarian agent. The results obtained from the rationality assumption could be adjusted to non-rational settings. On the contrary, the results obtained from specific treatments are good for these treatment settings only, which could only represent a subgroup of humans.
> >
> > I did inspect the code the authors provided during the review. The signaling function $H$ takes $\pi$ as one of the inputs. In simple games this $\pi$ could be represented by a vector or some similar objects. In this case a neural network $H$ could definitely generalize the vector to unseen vectors. In complicated tasks, where $\pi$ is, say, a neural network as well, I don't see how $H$ takes a neural network as an input and also generalizes that to unseen $\pi$ neural networks (not until the authors provide an implementation and experiments for that).
> >
> > My rating would remain the same primarily for the second issue.

---

> > > ### Author Response · Authors · 2023-08-16
> > >
> > > We thank the reviewer for the follow-up questions!
> > >
> > > **[Rationality assumption]**
> > >
> > > We want to first clarify that our framework is intended to incorporate a general set of human models, including the one with standard rationality assumption. As shown in our simulation and experiments, our approach can work with the standard rational models, the models from behavioral economics (e.g., TH-Model), and even models learned from data.  That is, we do not assume a new human model and work with that particular model. Instead, we are proposing a new framework that can work with a more general set of human models (including the standard one).  In this sense, we believe our work is a generalization of standard approaches that work with only rationality assumptions.
> > >
> > > We agree with the reviewer that the rationality assumption succinctly describes a fully utilitarian agent, and it provides an elegant formulation for us to develop mechanisms and algorithms with human participation.  In the meantime, we also want to highlight that this rationality assumption has often been criticized of not being able to accurately describe human behavior. The research field of behavioral economics has been dedicated to address this criticism. Moreover, there is a growing effort in combining ML and cognitive science to provide more accurate and interpretable human models that go beyond the standard rationality assumption (e.g., see Bourgin et al. ICML 2019 and Peterson et al. Science 2021 in our rebuttal).  Our work provides a framework to leverage these new developments of human models into the design of mechanisms in the context of information design.
> > >
> > > **[Scalability concern of human models]**
> > >
> > > The first point we want to mention is that our approach works with both close-form and data-driven forms of human models.  As shown in our human-subject experiments in Figure 4, if we have a reasonably accurate close-form representation of humans (e.g., TH-Model), we can achieve a reasonably good performance even without using data-driven human models.  The reviewer’s concern is mainly associated with the case when we aim to utilize data-driven forms.
> > >
> > > We now address the reviewer’s concern about having $\pi$ as the input in training the human model $H$ with a data-driven approach. Recall that $\pi(\sigma|\theta)$ is a conditional distribution specifying the probability of sending a signal $\sigma$ given a realized state $\theta$.  In cases when the space of signals and states is discrete, each information policy $\pi$ can be represented as a table (i.e., it can be represented as a vector), with each cell specifying the conditional probability. When training $H$, we only need to specify the input $\pi$ in this tabular representation.
> > >
> > > One note we want to clarify is that, in our HAIDNet framework, we have optimized a neural network for generating policies. This neural network encodes information policies for all problem instances within the given setting, i.e., it takes a problem instance (i.e., realization of priors, utilities) as input and outputs an information policy for that problem instance in the above tabular form.  When we are training $H$, our input is the problem instance with an information policy.  We do not need to feed the entire neural network as the input, we only need to provide one information policy as the input. So a tabular representation for $\pi$ is sufficient in training $H$ in our framework.
> > >
> > > The above discussion is under the settings with discrete states/signals. When we want to expand the setting to continuous setting (also raised by reviewers AWce and 5W54, and we have provided responses to them as well), it creates additional challenges. We should note that this challenge is not unique to our approach; it also exists in the traditional information design literature.  One potential method to address this is through discretization, though it generally requires some smoothness assumption (e.g., small deviation of states wouldn’t lead to significant different outcomes) to ensure the discretization error can be bounded.  Moreover, as mentioned in our rebuttal, the recent research of combining closed-form representations with data-driven approaches in developing human models could also mitigate this concern.
> > >
> > > Please let us know if you have additional questions.

---

> > > ### Author Response · Authors · 2023-08-19
> > > **Follow up to the Reviewer**
> > >
> > > We would like to thank the reviewer again for the comments. We believe we have addressed the reviewer’s concerns, including the major one:
> > >
> > > >“In complicated tasks, where $\pi$ is, perhaps, a neural network, I don't understand how $H$ accepts a neural network as an input and also generalizes that to unseen $\pi$ neural networks (not until the authors provide an implementation and experiments for that)”
> > >
> > > We believe the reviewer has a misunderstanding of our setting. As mentioned in our previous response, within our framework, $H$ takes instances of $\pi$ as inputs. These instances can be represented as vectors, even in complex tasks. Therefore, there is no need to input an entire neural network when training $H$, and our current implementation already takes care of encoding instances of $\pi$ as vectors.
> > >
> > > We hope this clarification addresses the reviewer's concern. Please let us know if there are any additional questions.

---

> > > > ### Comment · Reviewer_34E5 · 2023-08-21
> > > > **Response**
> > > >
> > > > I thank the authors for providing further responses. The responses make sense but they had been taken into consideration when the review was conducted. There are definitely ways to represent a policy into vectors, and ways to input one neural network into another. Nevertheless, these remain speculative in the context of information design. The implementation is up to simple environment and simple policies. This brings a weakness in an objective way.
> > > >
> > > > Regarding rationality, the work obviously did not assume any human model. However it works for only the model it was trained on. It's quite exploitative. To make the method work in real problems, one will need to further provide a human model that describes the group of humans they are trying to persuade. This isn't realistic. What if the human model is inaccurate? What if the group of humans are aware of your exploitation? The work does not provide an answer (not something that potentially solves it either).

---

> > > > > ### Author Response · Authors · 2023-08-21
> > > > >
> > > > > We thank the reviewer for engaging in the discussion.
> > > > >
> > > > > > "There are definitely ways to represent a policy into vectors, and ways to input one neural network into another. Nevertheless, these remain speculative in the context of information design."
> > > > >
> > > > > As in our previous response, we want to clarify one potential misunderstanding: we don’t need to encode instances of information policy as neural networks when providing them as inputs to train $H$. They can be encoded as vectors in our framework (this is always true in discrete settings, and can be approximated with discretization in continuous settings). Our current implementation has reflected this, and it is not merely “speculative”. We do agree with the scalability challenge; please see our response to Reviewer 5W54.
> > > > >
> > > > > > "What if the human model is inaccurate?”
> > > > >
> > > > > This is exactly one of the main motivations of this work. The standard rationality model in information design is often criticized for being an inaccurate human model in many real-world contexts, as highlighted in our intro. Moreover, as shown in our human-subject experiments (see Table 4 and Figure 4), models from behavioral economics (TH-Model) and data-driven models are better predictors of real-world human behavior and lead to significantly better sender payoff compared with the standard rationality model, i.e., the rationality model is the “inaccurate” model in our experiment with real human participants.
> > > > >
> > > > >
> > > > > Note that even in standard information design literature, we also need to provide a human model (the rationality model) for the design, i.e., the reviewer’s concern on inaccurate models holds for the information design literature, instead of just for our work. In fact, instead of being limited to the rationality model as in most of the literature, as demonstrated in our simulations and experiments, our framework can incorporate models from behavioral economics, data-driven models, or stick to the standard rationality models (our framework offers computational advantages in this case). That is, our framework is more general and gives the designer flexibility to incorporate the appropriate model to address the inaccurate model concern.
> > > > >
> > > > > > “What if the group of humans are aware of your exploitation?”
> > > > >
> > > > > We assume the reviewer is referring to the case that humans might adjust their behavior. Our work assumes human behavior stays the same during training and deployment of HAIDNet (e.g., in a stable environment).  We agree that it is an interesting and challenging future work to understand the dynamic of human behavior in a sequential learning setting (this problem aligns with the recent literature on strategic ML and performative prediction), but we consider it beyond the scope of this work.

---

> > > > > > ### Comment · Reviewer_34E5 · 2023-08-21
> > > > > > **Response**
> > > > > >
> > > > > > I thank the authors for their clarification.
> > > > > >
> > > > > > I agree with the authors that the framework has many potential things to achieve. At the moment, I have to evaluate the work mostly by the manuscript and the code, which represent works that are done. Hence I stick to my initial evaluation.

---

### Official Review · Reviewer_5W54 · 2023-07-09

**Soundness:** 4 excellent
**Presentation:** 3 good
**Contribution:** 2 fair
**Rating:** 6
**Confidence:** 4

**Summary:**

Information design is the problem in which a sender would like to optimize the information sent to a receiver, such that the receiver takes actions that the sender likes. As a simple example, a company may want to optimize what information they communicate about their product, to get consumers to buy the product. The typical Bayesian-rational model assumes that the receiver knows the policy by which the sender chooses what information to communicate, performs an appropriate Bayesian update, and then chooses an action that maximizes their expected utility (which can be different from the sender’s utility). The authors consider a more general setting in which the receiver is given the same information, but no constraints are placed on how they act on this information.

In general, information design problems are very difficult to solve (one result shows that a particular setting is #P-hard). The authors suggest that we instead solve information design problems using deep learning: in particular, we train a neural network (HAIDNet) to take in a specification of an information design problem, and output a policy for information to communicate. Crucially, HAIDNet takes in a (differentiable) specification of the receiver’s policy, and so can also work with models of the receiver that are not Bayesian-rational – including a neural network trained to mimic human behavior.

Experiments show that in small problems where the optimal policy is known, HAIDNet produces results that are nearly optimal (achieve a sender reward that is single-digit percentages away from optimal, well above that achieved by a random policy). HAIDNet continues to produce good-looking results in larger settings where the optimal policy is not known, though it is hard to tell how good the results are. Finally, the authors run an experiment with real humans, where they first elicit the human policy, train a neural network to imitate the humans, and then train HAIDNet to solve the information design problem with the neural network as the receiver policy. They show that HAIDNet produces a solution that, when tested with real humans, significantly outperforms the optimal solutions under the assumption of Boltzmann rationality, and non-significantly outperforms the TH-model (a model from prior literature), illustrating the benefit of using data-driven models of human behavior.

**Strengths:**

1. By far the biggest strength of this paper is its experiment with humans. Showing strong performance with real humans is a significantly higher bar than showing good results in simulations with human stand-ins; this paper meets this higher bar.
2. The setting of information design is scientifically interesting, and to my knowledge deep learning has not been applied to it before (though I am not familiar with the literature).
3. The authors show two major benefits from using deep learning: the ability to approximate answers with arbitrary (differentiable) human behavior models, as well as the ability to compute approximate solutions much faster than with e.g. linear programming. (That being said, I do not know the classical state of the art for approximately solving information design problems.)
4. The authors’ approach is simple and natural, suggesting it would be easy to replicate.

**Weaknesses:**

1. Scalability: Information design problems require networks to take full policies as an input to the network. This will be a challenge for large problem instances, and isn’t tackled in this paper.
2. Ethics: Encoding realistic models of human behavior into information design seems particularly likely to have negative social impact.

**Scalability**

Information design problems require networks to take full policies as an input to the network: in particular, the network modeling human behavior must take the full stochastic policy that the sender uses to make decisions as an input. (The HAIDNet does not need to take the full human policy as input, it just samples from the policy on appropriate inputs as needed.)

This can be done when there are a small number of possible states and possible signals to send, as in the simple experiments done in this paper. However, in realistic settings such as recommender engines deciding which ratings to display, or politicians deciding how to design policy experiments, there will be a huge number of possible states and signals, which cannot even be enumerated. It’s not clear how HAIDNet would scale to these situations.

(That being said, it does not look impossible, or even necessarily very hard. Ultimately there is a bi-level optimization problem, and while those are hard to solve, there are many techniques that get traction on the problem.)

I do think the contribution of the paper is significantly more limited due to the unscalable nature of the current architecture.

**Ethics**

I am worried that this research will primarily contribute to negative impacts on society, and have flagged it for ethics review below. If I were making the decision based only on the information I currently have, I would not publish the paper and would encourage the authors to set aside this research area and move on to other work.

However, this is not one of the areas mentioned in the NeurIPS code of ethics, so I am not incorporating this into my evaluation for the paper, which I will base just on the scientific merits of the paper. I will leave it to the area chair / ethics reviewers to decide how to incorporate ethical considerations into the decision.

The authors note that typically in information design, the sender is the party in power, while the receiver is not, and so optimizing for the sender’s utility can result in using a sender’s informational advantage to exploit the receivers. I agree with this, but in addition I think moving from a Bayesian-rational model to a data-driven human behavioral model makes the problem much worse

For a Bayesian-rational receiver, we have the following property: the receiver’s beliefs over the underlying state are well-calibrated (they follow from an appropriate Bayesian update that incorporates how the information was generated). Relative to getting no signal at all from the sender, this can never harm the receiver (in expectation), since for a rational Bayesian agent the value of information is always positive. This bounds the amount of harm that can be done from information design.

(It is of course still possible that the receiver does much worse than if the sender simply provided the information most useful to the receiver, which we would count as a harm in many cases, e.g. the sender may probabilistically choose not to reveal that a more expensive product is actually lower quality, so that the receiver buys the more expensive product some of the time; we would plausibly consider this unethical since the sender could have provided better quality information allowing the receiver to make a more informed decision.)

In reality, human receivers will not be Bayesian-rational. However, a sender policy that was chosen based on a Bayesian-rational receiver model is still going to produce actions whose intended effects are to give the receiver true information. In particular, it will _not_ choose actions based on their propensity to mislead or trick the receiver, because it is impossible to trick the Bayesian-rational receiver.

However, once we move to data-driven models of human behavior, we lose these guarantees. A data-driven model of human behavior will likely also include many human biases, and so a sender policy optimized against such a data-driven model will learn to exploit these biases. For example, it may:

1. Appeal to irrational fears (e.g. selling products by making some risks emotionally salient, even if the risks are tiny in practice – maybe an expensive shark repellant to ward off shark attacks)
2. Understand which claims humans can check (i.e. which aspects of the sender’s policy the receiver can observe and is sensitive to), and lie about any claims that can’t be easily checked
3. Identify markers of credibility, and design signals that exploit credibility marker (e.g. endorsements from celebrities or scientific experts with strong public reputations, but who give their endorsement freely to even poor quality products).

I’m sure there are many more such applications, these are just for illustration. We already see some of these effects with advertising and politics, and that is with relatively unsophisticated models of human behavior.

All of these effects could be significantly exacerbated if there are significantly more accurate data-driven models of human behaviors, or if these models could be personalized to specific individuals. In the case of personalized models, I would also expect to see significant inequality effects, where more elite, well-connected, and knowledgeable people will better understand how automated information design results in manipulation, and will be better able to counteract its effects, and meanwhile worse-off people will be more vulnerable to manipulation.

There are of course potential positive applications as well: for example, this could be used to design more effective campaigns to improve public health (e.g. leveraging knowledge of human biases to get people to wash their hands more often). However, it seems like the negative applications are both more numerous and more likely to happen.

**Questions:**

Do you have any plans or ideas for scaling up HAIDNet to much larger problems, where the full state space is implicit and the receiver policies cannot be enumerated in full?

Can you elaborate more on the ethical implications / broader impacts of this research? (See discussion above; in particular I’m interested in a response to the point about exploiting systematic human biases.)

Minor suggestions:

Line 305: Mention that in the human experiment you tell the humans the prior in addition to the conditional probabilities (currently it only mentions the conditional probabilities).

**Limitations:**

The authors have some discussion of potential negative societal impacts, but I think it is insufficient – see Weaknesses.

I would also recommend that the authors briefly discuss the limitation of scalability.

To get additional space, I think it would be fine to remove the discussion about how gradient descent does not find optimal solutions, and how the resulting solutions may not be robust to distributional shift: the NeurIPS audience will tend to assume these two limitations by default for any paper applying deep learning, unless the paper explicitly claims otherwise.

---

> ### Author Rebuttal · Authors · 2023-08-10
>
> Thanks for the insightful comments! We will incorporate the clarification suggestions. Below we respond to the two major comments.
>
> **[Scalability]**
>
> In our framework, there are two separate scalability considerations:
>
> *Scalability of optimizing information policy*: We'd like to begin by noting that scalability is a recognized challenge in the traditional study of information design. A key motivation for our work is the result by Dughmi and Xu [15] that computing the optimal information policy is #P-hard.
>
> We have empirically examined the scalability of our approach in Appendix B.1.2. Specifically, as depicted in Table 5, for the more computational heavy setting with multiple receivers, the standard linear programming (LP) method of solving the information policy scales poorly, taking 0.3 seconds to solve an instance with one receiver and over 23 hours to address an instance with 17 receivers. In contrast, our approach demonstrates better empirical scalability. It takes 0.3 hours to train a neural network (NN) for one receiver and 1.12 hours for 17 receivers. It's noteworthy that the trained NN can generate information policies to all problem instances, and the testing time to generate a policy for an instance takes less than 0.352 seconds, even for cases with 17 receivers. These empirical results suggest that our method offers a much more scalable solution for designing information policies compared to traditional methods.
>
> In scenarios with continuous action/state spaces, the information design problem is generally challenging to solve, both in the traditional information design literature and in our approach. To address this challenge, one potential approach is to employ discretization. However, this approach requires some smoothness assumption (e.g., small deviation of actions wouldn’t lead to significant different outcomes) to ensure the discretization error can be bounded.
>
> We will add discussion in the main text in the revision.
>
> *Scalability of approximating human behavior*:  Our framework is designed to accommodate various representations of human behavior. As shown in our evaluations, it effectively handles both the standard closed-form representations of human models (e.g., Bayesian rationality or TH-model) and data-driven forms. While we concur with the reviewer's concern regarding the scalability of training a purely data-driven human model, recent studies at the intersection of cognitive science and ML have shown that we can utilize established human models (e.g., Bayesian rationality or the TH-model) as priors and harness data to refine these models [Bourgin et al. ICML 2019, Peterson et al. Science 2021]. The main benefit of this approach is that it provides a reasonable model to begin with even when we do not have enough data and could lead to a more accurate model with more data. Our framework can seamlessly integrate with human models developed using such methodologies, mitigating the scalability concerns.
>
> - Bourgin et al. "Cognitive model priors for predicting human decisions." ICML. 2019.
> - Peterson et al. "Using large-scale experiments and machine learning to discover theories of human decision-making." Science. 2021.
>
> **[Ethics]**
>
> We wholeheartedly agree with the reviewer’s concern, which is the reason we brought up this point during our discussion. The information design literature, and more broadly, the principal-agent problem or Stackelberg game in economics, presents a scenario where one advantaged party might exploit its informational advantage over the disadvantaged party. As the reviewer noted, this concern could be amplified as we acquire more accurate knowledge about the disadvantaged party.
>
> That said, we still believe it is important to conduct our research. First, as also pointed out by the reviewer, this concern of exploiting irrational human behavior is ubiquitous and is already being deployed by private sectors in the domain of marketing or political campaigns. In order to develop risk mitigation methods, it is our belief that we first need to understand the capacities of this approach. Secondly, there are also benign ways of utilizing this approach, e.g., encouraging the general public to take actions promoting social good or preventing individuals from taking self-sabotaging decisions. Without a public grasp of both the potential risks and benefits, we risk leaving the utilization of these techniques to the private sectors exclusively, which, arguably, could present an even more concerning situation.
>
> We also want to discuss two potential risk mitigation methods in light of the concern raised. Firstly, on the technical front, we might employ differential privacy techniques to control the amount of personal data used in training human behavior models. Differential privacy offers a means to balance privacy with utility, typically by introducing controlled noise into the data. This mechanism is potentially helpful in mitigating the exploitation of marginalized groups, an issue raised by the ethics reviewer. Secondly, from the policy perspective, after we develop a deep understanding of the capability of information design with data-driven human models, we as a society could and should discuss the tradeoff of the utility we can gain from this method and the harm we might suffer. This tradeoff discussion can then lead to the developments of regulations and policies. For example, we could add constraints requiring that the deployed information policy shouldn’t lead to much lower receiver utility, compared with deploying information policy designed assuming Bayesian rational models. This is similar to one common approach of imposing fairness constraints in the recent fairness literature. But again, these discussions requires us to develop a good understanding of the capability of this approach in the first place
>
> We will provide more discussion on the above in the revision.

---

> > ### Comment · Reviewer_5W54 · 2023-08-12
> > **Thanks for the response**
> >
> > I have read the authors' rebuttal and the other reviews, and am maintaining my score of 6 (again noting that it doesn't take into account the ethics flag).
> >
> > ## Scalability
> >
> > Yes, I agree with everything you write here. I was critiquing the scalability to larger games in particular (whether continuous state/actions, or discrete states/actions with a large number of states + actions). I agree that the method can already take advantage of other advances in human modeling, and also agree that the method already shows significant advantages relative to exact solvers.
> >
> > ## Ethics
> >
> > While I agree that it is helpful to understand a problem in order to fix it, from this perspective I think the natural thing to do is to investigate the existing techniques that people use, and attempt to design risk mitigation methods. I would not be designing novel techniques: if these techniques are already used then my time would be wasted reinventing the wheel; if the techniques are not used then I have just increased capacity for harm before I’ve even started to design mitigations.
> >
> > I agree there are also beneficial uses, but my expectation is that the harmful uses are more common and more impactful than the beneficial ones (though of course this is based on guesswork, and would benefit from a more thorough investigation). If the beneficial uses significantly outweighed the harmful uses I would withdraw my objection.

---

### Official Review · Reviewer_7rN5 · 2023-07-10

**Soundness:** 3 good
**Presentation:** 3 good
**Contribution:** 2 fair
**Rating:** 6
**Confidence:** 4

**Summary:**

The paper focuses on the problem of automated information design, where the sender strategically reveals information to persuade the receiver to take specific actions. The main contribution of this paper is addressing the challenge of modeling human behavior when individuals do not act as Bayesian rational agents. The problem is formalized as an optimization problem, and the authors propose a neural network architecture (HAIDNet) to solve this optimization problem. The effectiveness of their approach is evaluated through experiments conducted in both simulation and real world settings. In their simulations, they show that HAIDNet can solve settings _without_ efficient solutions, including multiple receivers and non-Bayesian receivers. They then fit a neural network model to model how human subjects update their preferences in practice, and show that HAIDNet significantly improves on baselines in convincing human subjects to change their behavior.

**Strengths:**

* The method proposed in the paper is intuitive and make sense.
* The writing is clear and effectively communicates the ideas, with Figure 1 providing a helpful visual illustration of the concepts.
* The experimental analysis is extensive and noteworthy, particularly with the inclusion of results with real human subjects. This adds credibility to the findings.

**Weaknesses:**

* It is unclear how scalable this approach is. To fit the neural network, a large amount of data needs to be collected, and such data is often scarce. For example, in their experiments, they used 2000 worker-rounds of product purchasing decisions to fit a small 3-layer neural network on a binary tasks , more realistic human models
* The proposed method requires differentiable human models, which can be hard to come by in practice, especially in cases where the human models are e.g. designed by hand or feature random components.
* The learned human models and information policies are hard to interpret, which is compounded by the issue of scarce data. It's not clear what exactly the small neural networks have learned in their experiments, and the authors do not discuss this in their paper. In practice, it seems important to

**Questions:**

1. I'd be interested in hearing more about the relationship between this work and Reddy, Levine, and Dragan 2021*. In that work, they also fit a neural network policy (albeit a larger, recurrent one) to synthesize observations to influence the actions of a human users. Similarly, I think there's some amount of related work in the human-assistance literature in general that feature similar algorithms (where a blackbox neural network is used to learn a policy that performs well according to a utility function with human subjects).
2. I'm somewhat puzzled about why you call it an architecture, when HAIDNet seems to me to be more of an algorithm for training arbitrary neural network architectures. In my case, I was a bit confused by this, and expected a special neural network architecture more directly designed for modeling information policies, but instead the paper uses 3-layer MLPs. I'd appreciate if this was made more clear in the introduction and in section 3.2.
3. Similarly, I would appreciate more discussion on the core assumptions of the HAIDNet framework. What do you think are the crucial parts of the algorithm related to the key insights, and which parts are contingent on implementation details? For example, is it important that you optimize the policy with SGD instead of an RL algorithm like PPO or DDPG? Is it important that the neural network is a 3-layer MLP?

\* Reddy, S., Levine, S. and Dragan, A., 2021, October. Assisted perception: optimizing observations to communicate state. In Conference on Robot Learning (pp. 748-764). PMLR.

**Limitations:**

The authors describe the key limitations of HAIDNet as 1) possibly finding only locally optimal solutions as it uses Adam (a local first-order optimizer) to update the information policy and 2) only getting good performance _in expectation_, as opposed to uniformly across all cases. However, I think in comparison to these two limitations, the scalability, differentiability, and interpretability issues described in the weaknesses section above are more important for using their method in practice.

**(Negative) Social impacts**
The authors correctly note that their method (as with most other work in this space) can be used for disinformation or user manipulation.

---

> ### Author Rebuttal · Authors · 2023-08-10
>
> Thanks for the valuable feedback and comments!
>
> **[Scalability]** Please see our responses to Reviewer 5W54.
>
> **[Differentiable human models]** While the standard rationality model is not differentiable, many of the other models are, e.g., discrete choice models and data-driven models. The differentiable assumption essentially requires human behavior to be smooth, which is often approximately satisfied if we consider the stochasticity of human behavior or focus on dealing with a population of humans.  In cases when the human model is not differentiable, we can utilize standard approaches, e.g., softmax relaxations, to approximate human behavior. We will add discussion in the revision.
>
> **[Interpretability]** For the interpretability of information policy, we would like to first note that this is an open question even in the standard information design literature. While in some settings, e.g., when the receiver’s optimal action depends only on the expected state, an interpretable optimal information policy can be characterized with certain mild assumptions (see, e.g., Arieli et al EC’20; Kolotilin et al, Theoretical Economics'22), to the best of our knowledge, there are no existing characterizations on interpretable information policy for general settings.  In our work, to increase interpretability, one potential approach is to add constraints in the sender’s optimization problem to force the information policy to satisfy certain interpretability conditions. We will add more discussions on this point in the revision.
>
> For the interpretability of human models, we want to first highlight that our framework is designed to accommodate various representations of human behavior, including the standard closed-form representations of human models (e.g., Bayesian rationality or TH-model) and data-driven forms. While the former is interpretable, the latter is not in general. Developing an accurate and interpretable human model is actually an open and ongoing research question at the intersection of cognitive science and ML (e.g., Bourgin et al. ICML 2019 and Peterson et al. Science 2021). Our framework can seamlessly integrate with human models developed using such methodologies, mitigating the interpretability concerns.
>
> - Bourgin et al. "Cognitive model priors for predicting human decisions." ICML.  2019.
> - Peterson et al. "Using large-scale experiments and machine learning to discover theories of human decision-making." Science. 2021.
>
> **[Comparison to prior work]** As discussed in the expanded related work in the appendix, on a conceptual level, our work is related to the growing attention in understanding, modeling, and accounting for human behavior in computation systems, especially in the context of human-robot or human-AI interactions. We have cited several works from the same lab of the work mentioned by the reviewer on human-robot interactions, and we will include this one as well. More specifically, our works are similar in the sense of aiming to develop realistic human models and design AI, ML, or the environment to work with humans. However, depending on the problem setting, the format of interactions and objectives are different for different works in this literature. These differences result different problem formulations and challenges, including the need to abstract proper forms of human bebavior, identifying the objectives to be optimized with data-driven approaches, etc.
>
> **[Architecture]** We called HAIDNet an architecture to highlight that we have included human models in the design. However, we agree with the reviewer that it might confuse readers with the architecture of neural networks. We will call it a framework in our revision.
>
> **[Assumptions]** Thanks for the suggestions. We will more explicitly list the assumptions. In particular, we require two assumptions: (1) the behavior of humans used for training HAIDNet is draw from the same distribution of the human behavior during deployment (i.e., the standard training/test assumption in ML), and (2) human models are differentiable (this assumption is soft, as it can be approximated using soft-max relaxations if it’s not).
>
> **[Importance of implementation details]** Our goal is to show that a data-driven approach can be used to find a near-optimal information policy for general human receiver models. Our investigation has adopted the most standard setup: Fully-connected MLP + SGD, and this setup already shows promise in improving the outcomes. We did not examine the other architecture and approaches but agree it would be interesting future work.
>
> **[Expanding discussion]** Thanks for the suggestions. We agree with the reviewer’s comment and will discuss the scalability, differentiability, and interpretability in our revision.

---

> > ### Comment · Reviewer_7rN5 · 2023-08-15
> > **Thanks for the response!**
> >
> > Thanks to the authors for the response, especially with respect to my concerns on scalability (included in the response to reviewer 5W54) and interpretability.
> >
> > I've increased my score to a 6 in response.

---

### Official Review · Reviewer_AWce · 2023-07-24

**Soundness:** 3 good
**Presentation:** 3 good
**Contribution:** 3 good
**Rating:** 5
**Confidence:** 1

**Summary:**

This paper proposes a neural network based framework for automated information design that can optimise both the senders and the receivers behaviour. In situations where the receivers behaviour cannot be approximated analytically, a neural network that learns the preferences from existing data can be utilised. In all situations the sender's behaviour is optimised via minimising a differentiable loss. In experiments they show that this framework can closely approximate the optimal solution in smaller scale problems, and can also be utilised with real human participants where the receivers behaviour must be learnt, and outperforms the less data-driven baselines.

**Strengths:**

As somebody not familiar with the field of automated information design, I still enjoyed reading the paper.
I liked the presentation of the motivations and the preliminaries on Bayesian persuasion.
Relaxing the assumption of Bayesian rationality and demonstrating that a more data-driven approach (backed by ML) can be utilised is hugely useful, especially since it is demonstrated on a study with real human participants.

That a learning based approach (utilising a small MLP) can be trained significantly quicker and still recover good approximations of optimal solutions is a big plus for the method (provided the practitioner has the appropriate hardware to train a neural network fast).

**Weaknesses:**

(Again not very familiar with the field) I am struggling a little bit to understand the exact contributions of this paper.
Proposing to use a neural network and that machinery to solve Eqn 2 doesn't seem to be novel (as mentioned in the paper lines 168-169).
There does seem to be in attempting to learn human behaviour in these scenarios from data with a neural network.
I am a little unsure as to whether there have been ML (specifically neural network based) approaches to this problem in this field.
Nonetheless, since the ML setup+framing is such a crucial part of the proposed method, the paper would benefit from a more in depth description and discussion of the different components.
1. What exactly is the input to the neural network, what size is it (it looks to be a table that is presumably flattened?), how does that scale wrt problem instances.
2. What architectures were considered/studied for this particular problem.
3. How can this approach scale when a tabular representation doesn't suffice.

Related to the previous point, from my understanding of the paper a big contribution is that human behaviour in these tasks is much better approximated with a learning based approach. Thus, how exactly that module is setup is very important and should be discussed in more detail.
In particular, how exactly the sender's information is policy is represented (and how this might or might not scale). In addition, the human's behaviour is approximated using existing data. However, as soon as the sender's policy changes to something that has not already been observed then we are reliant on the generalisation of the neural network to accurately capture the human's behaviour. How well this part of the framework generalises to the optimised sender policies should also be looked at.

I like Figure 3, testing the generalisation. However, I would like a more absolute measure of the performance as well. Does the best
performing HAIDNet policy actually perform well on the task?
Additionall,y I would like more discussion on these results. Particularly for cases like $\beta_H = 100$, where it seems to 'generalise' much better than say $\beta_H=1$. What is the difference in the learned policies? Why for $\beta_H=\infty$ does the performance dip around 20?

I like Table 4 a lot, some more discussion on it would be good to really drive home the differences between the Bayesian rational assumption and real observed human behaviour. Also a comparision between the NN and TH-Model to try and identify what about the TH-Model is preventing it from learning a better solution. Also, where does the neural network make errors? It's performance is indeed higher than the others, but there is still quite a bit of headroom.
Also for Figure 4, a more qualitative discussion on the resulting policies learned by the methods.

Minor:

> "HAIDNet essentially recovers the optimal information policy in almost all scenarios"
Would be nice to quantify this a bit more, since there do look to be small differences in Figure 2.

You mention a boosting and aggregation approach is taken, please described this in more detail.

Is the information in C.1 required/does it need to be reported in the paper (Race/Ethnicity in particular)?

**Questions:**

(I have put my important questions in the section above - apologies)


**Limitations:**

Yes, limitations and potential negative societal implications have been discussed.

---

> ### Author Rebuttal · Authors · 2023-08-10
>
> We thank the reviewer for the thoughtful review.
>
> **[Contributions]** We first highlight our contributions. Firstly, we introduced a data-driven optimization framework for information design. Note that even in standard settings where humans are Bayesian rational, designing optimal information policy is known to be #P hard. Our proposed approach can more efficiently identify near-optimal information policies compared with standard techniques. Secondly, our approach can accommodate different representations of human behavior beyond standard assumption of Bayesian rationality. This flexibility also makes it straightforward to incorporate recent research trends that combine domain knowledge with data-driven models for human behavior. Lastly, we have validated our methods through human-subject experiments, showcasing the practical viability of our approaches.
>
> Below we respond to your other comments/questions. We will incorporate your suggestions and the following clarifications in the revision.
>
> **[Scalability]** Please see our responses to Reviewer 5W54.
>
> **[Network input and continuous input space]** The input to the neural network (NN) specifies the problem's specifications: the prior distribution, sender utility, and receiver utility. For instance, in a single-receiver setting with $M$ states and $N$ actions, the input size is given by $M * (2N+1)$. In scenarios with continuous action/state spaces (i.e., the input can't be tabulated), the information design problem is generally challenging to solve, both in the traditional information design literature and in our approach. To address this challenge, one common approach is to employ discretization. However, this approach requires some smoothness assumption (e.g., small deviation of actions wouldn’t lead to significant different outcomes) to ensure the discretization error can be bounded.
>
> **[Network architecture]** As described in Section 3.2 and Appendix B.3, we utilized a 3-layer fully connected neural network with ReLU activation functions. We want to emphasize that the objective of our study is to demonstrate the viability of a data-driven approach for automated information design. We are not positing that our chosen architecture is the best possible. Nevertheless, even with this relatively standard choice, our results have already been promising.
>
> **[Receiver modeling without representative data]** We want to highlight that our framework is designed to accommodate various representations of human behavior. As shown in our evaluations, it can handle both standard closed-form representations of human models (e.g., Bayesian rationality or TH-model) and data-driven forms (e.g., neural networks). The reviewer's concern regarding the challenge of developing human models when data is not representative (e.g., encountering situations not in the historical data) pertains specifically to data-driven human models. Indeed, if our human model is entirely data-driven, we would require the training data to be representative to ensure generalizability, as is commonly required in supervised learning.
>
> That said, the reviewer’s concern can be mitigated by incorporating the recent studies in utilizing existing close-form human models as priors and leverage behavioral data to improve the models [Bourgin et al. ICML 2019, Peterson et al. Science 2021]. The main benefit of this approach is that in cases when the data is insufficient, the model behavior would fall back to existing human models. Our framework can seamlessly integrate with human models developed using such methodologies.
>
> - Bourgin et al. "Cognitive model priors for predicting human decisions." ICML.  2019.
> - Peterson et al. "Using large-scale experiments and machine learning to discover theories of human decision-making." Science. 2021.
>
> **[Explain Figure 3]**  The parameter $\beta_H$​ in this human model represents the stochasticity level of human behavior. A smaller value of $\beta_H$​ indicates more random human behavior. Consequently, human models with $\beta_H=1$ depict behavior that is close to random. As demonstrated in the left-most column of Figure 3, in such scenarios, all information policies lead to similar outcomes. Conversely, when $\beta_H$ approaches infinity, human behavior becomes deterministic. In these instances, an accurate understanding of human behavior is essential for effective performance, as illustrated in the rightmost column of Figure 3. To explain the relative performance dips observed around $\beta_H=20$ for policies trained on the human model with $\beta_H=\infty$: For smaller values of $\beta_H$, the performance across different policies tends to be similar. However, for considerably large values of $\beta_H$, a $\beta_H$ value of infinity serves as a reasonable approximation. Thus, the most significant relative performance gap for a policy trained on $\beta_H=\infty$ occurs with moderate values of $\beta_H$.
>
> **[Test accuracy of human models]** In our human-subject experiments, we are learning a model to predict the behavior for a population of humans. Together with the intrinsic stochasticity of human behavior, it is not possible to reach perfect accuracy in predicting human behavior. In fact, Tang and Ho [56] have estimated that close to 20% of worker behavior in a similar setting is random.
>
> **[Optimization procedure]** The boosting procedure is only used in settings with Bayesian rational receivers. Since this human model is not differentiable, we adopt the common practice of using differentiable softmax function to approximate its behavior, which leads to higher errors. We address this by taking a boosting approach of iteratively training networks, reweighting data distributions, and aggregate learned neural networks to reduce the error.
>
> **[Worker demographics]** It is a common practice to report the demographics of the user studies, for transparency and generalizability reasons.

---

> > ### Comment · Reviewer_AWce · 2023-08-16
> > **Thanks for your reply**
> >
> > Reading the other reviews and your replies, most of my concerns have been somewhat alleviated and I will raise my score.
> > Please do incorporate some of your replies into the main paper, particularly that the focus is not on a particular neural network architecture, and that the particular architecture and input representation you used would struggle to scale to larger and more complex problem domains (and would this be an avenue for future work).

---

> > > ### Author Response · Authors · 2023-08-17
> > >
> > > Thanks! Yes, we will definitely incorporate the discussion in our revisions. We appreciate all the feedback that helps us improve the paper.

---

### Decision · Program_Chairs · 2023-09-21

**Decision:**

Accept (poster)

**Comment:**

The reviews and discussion give mixed support, acknowledging the technical contributions but also raising potential ethical concerns. We encourage the authors to use the feedback to revise and strengthen the technical and ethical discussions in this work.